# Provably Efficient Algorithm for Best Scoring Rule Identification in Online Principal-Agent Information Acquisition

Zichen Wang [1]   Chuanhao Li [2]   Huazheng Wang [3]

## Abstract

We investigate the problem of identifying the optimal scoring rule within the principal-agent framework for online information acquisition problem. We focus on the principal's perspective, seeking to determine the desired scoring rule through interactions with the agent. To address this challenge, we propose two algorithms: OIAFC and OIAFB, tailored for fixed confidence and fixed budget settings, respectively. Our theoretical analysis demonstrates that OIAFC can extract the desired $(\epsilon, \delta)$-scoring rule with a efficient instance-dependent sample complexity or an instance-independent sample complexity. Our analysis also shows that OIAFB matches the instance-independent performance bound of OIAFC, while both algorithms share the same complexity across fixed confidence and fixed budget settings.

## 1. Introduction

The information acquisition problem, framed within the framework of the principal-agent model (Laffont & Maskin, 1981), studies how a principal hires an agent to gather critical information for decision-making processes (Oesterheld & Conitzer, 2020; Neyman et al., 2021). In this setup, the principal relies on the effort and expertise of the agent to provide high-quality information. Consider a strategic interaction between a manager (principal) and a domain expert (agent). The manager is tasked with evaluating the suitability of an advertising strategy for a given market. The effectiveness of this strategy depends on an uncertain underlying factor, which we refer to as the environment state $\omega$. This state could represent characteristics of the market,

such as consumer preferences or competitive dynamics, that influence the strategy's potential. The manager first designs a scoring rule $S$ and announces it to the expert. The expert investigates $\omega$ by selecting a research method (action $k$), which incurs a cost $c_k$. The expert chooses the research method that maximizes its expected profit, i.e., the method that maximizes $S(\hat{\sigma}, \omega) - c_k$, where $S(\hat{\sigma}, \omega)$ represents the payment from the manager based on the accuracy of the report. After performing the selected method, the expert observes the outcome $o$ and produces a report $\hat{\sigma}$. The manager then uses $\hat{\sigma}$ to make an investment decision $d$ (e.g., implementing the proposed strategy). The utility of the manager $u(\omega, d)$ is determined based on the true state $\omega$ and $d$.

While most existing research (Neyman et al., 2021; Li et al., 2022; Hartline et al., 2023) focuses on designing optimal scoring rules in offline settings, relatively little attention has been given to the online scenario, where the principal seeks to achieve learning objectives through repeated interactions with the agent. To the best of our knowledge, only two prior studies have explored the online information acquisition problem (Chen et al., 2023; Cacciamani et al., 2023). However, these studies primarily focus on regret minimization from the principal's perspective, aiming to reduce cumulative regret from sub-optimal decisions. A less-explored aspect is best scoring rule identification (BSRI), where the principal's goal is to identify an estimated optimal scoring rule that maximizes expected profit through repeated interactions.

Chen et al. (2023) presents instance-independent BSRI results for their algorithm in the fixed-confidence setting (see Corollary 4.4 in Chen et al. (2023)). They show that their algorithm can identify an $(\epsilon, \delta)$-optimal scoring rule using $\tilde{O}(C_{\mathcal{O}}^3 K^6 \epsilon^{-3})$ samples, where $K$ represents the number of the agent's actions, and $C_{\mathcal{O}}$ denotes the total number of possible observations. However, this sample complexity is relatively higher than that of standard best arm identification algorithms for multi-armed bandit problems (Even-Dar et al., 2002; Kalyanakrishnan et al., 2012), which typically require only $\tilde{O}(K\epsilon^{-2})$ samples. Moreover, no instance-dependent results are currently available for fixed-confidence BSRI, where the sample complexity upper bound would depend on the expected profit gap between the optimal and sub-optimal

[1]Department of Electrical and Computer Engineering and Coordinated Science Laboratory, University of Illinois Urbana-Champaign, Illinois, USA [2]Department of Statistics and Data Science, Yale University, Connecticut, USA [3]School of Electronic Engineering and Computer Science, Oregon State University, USA. Correspondence to: Zichen Wang <zichenw6@illinois.edu>.

*Proceedings of the 42$^{nd}$ International Conference on Machine Learning*, Vancouver, Canada. PMLR 267, 2025. Copyright 2025 by the author(s).

actions (research methods). Additionally, the fixed-budget setting of BSRI also remains unexplored, leaving open questions about performance in scenarios with limited sample availability.

Based on the above observation, we address the BSRI problem and build upon the setting and results presented in Chen et al. (2023). We introduce two algorithms: OIAFC for the fixed-confidence setting and OIAFB for the fixed-budget setting. We introduce new trade-off parameters $\alpha_t^k$, along with a breaking rule and parameter $\beta_t$, as well as a decision rule. These innovations ensure that we achieve a $(\epsilon, \delta)$-optimal scoring rule while maintaining an instance-dependent sample complexity upper bound. To the best of our knowledge, our results are the first instance-dependent results for the online information acquisition problem. The contributions are summarized as follows:

- OIAFC identifies the $(\epsilon, \delta)$-optimal scoring rule with an instance-dependent sample complexity upper bound of $\tilde{O}\left(M\left(\sum_{k \neq k^*} \Delta_k^{-2}\right) + \epsilon^{-2}\right)$ in the fixed-confidence setting, where $\Delta_k$ represents the profit gap, and $M \leq K \times C_{\mathcal{O}}$. We also demonstrate that in a simple setting, OIAFC's instance-dependent result aligns with the known instance-dependent outcomes for the fixed-confidence best-arm identification problem in multi-armed bandits.

- OIAFC identifies the $(\epsilon, \delta)$-optimal scoring rule with an instance-independent sample complexity upper bound of $\tilde{O}(MK\epsilon^{-2})$ in the fixed-confidence setting, which improves the result proposed by Chen et al. (2023). This result aligns with the known instance-independent result for the fixed-confidence best arm identification problem in the MAB.

- The OIAFB algorithm, consistent with the instance-independent results of OIAFC, identifies an $(\epsilon, \delta)$-optimal scoring rule, given a budget of $T = \tilde{O}(MK\epsilon^{-2})$.

## 2. Related Work

**Best arm/policy identification.** Best arm identification in bandits (Mannor & Tsitsiklis, 2004; Even-Dar et al., 2006; Bubeck et al., 2009; Gabillon et al., 2011; Kalyanakrishnan et al., 2012; Gabillon et al., 2012; Jamieson et al., 2013; Garivier & Kaufmann, 2016; Chen et al., 2016; Wang et al., 2023; Azize et al., 2024) shares a similar goal with our problem: finding a near-optimal arm (scoring rule) with minimal sample complexity. The smaller the sample complexity, the better the algorithm's performance. However, one of the key differences between our setting and theirs is that we cannot directly pull an arm (Chen et al., 2023; Cacciamani et al., 2023); instead, we must use the scoring rule to incentivize

the agent to pull the desired arm. This complication requires us to learn the response region of arm $k$ (will be introduce in the following sections).

**Online learning in strategic environments.** Our problem falls under the topic of online learning in strategic environments. In this category, the online learner's reward in each round is influenced by their actions (such as the scoring rule in our paper) as well as the strategic responses of other players (such as the agent in our paper) in repeated games. These repeated games are grounded in influential economic models, including Stackelberg games (Sessa et al., 2020; Balcan et al., 2015; Haghtalab et al., 2022), auction design (Amin et al., 2013; Feng et al., 2017; Golrezaei et al., 2018; Guo et al., 2022), matching (Jagadeesan et al., 2021), contract design (Ho et al., 2014; Cohen et al., 2022; Zhu et al., 2022; Wu et al., 2024; Zuo, 2024a;b), Bayesian persuasion (Castiglioni et al., 2020; 2022; 2021; Bernasconi et al., 2022; 2023; Gan et al., 2021; Zu et al., 2021; Wu et al., 2022; Agrawal et al., 2024; Bacchiocchi et al., 2024), principal-agent games (Alon et al., 2021; Han et al., 2024; Ben-Porat et al., 2024; Ivanov et al., 2024; Scheid et al., 2024; Chakraborty et al.). Chen et al. (2023) presented the first study on online information acquisition, primarily focusing on the regret minimization problem. They also proposed a fixed-confidence result for best scoring rule identification, but their sample complexity upper bound is loose and instance independent.

**Best scoring rule.** Our problem is also related to the topic of offline optimal scoring rule design. Previous studies have made different assumptions regarding the number of states and action numbers. Some prior works ((Neyman et al., 2021; Hartline et al., 2023)) assume there are only two possible states. In contrast, our paper allows the number of states to be any finite number $\geq 2$. Similarly, some previous studies ((Chen & Yu, 2021; Li et al., 2022; Papireddygari & Waggoner, 2022)) assume the action number is $K = 2$. In our work, we allow $K$ to be any finite number $\geq 2$. Additionally, while all the papers mentioned above model the state as exogenously given, our paper allows for the prior of the state to potentially be influenced by the agent's actions (Chen et al., 2023).

## 3. Preliminaries

**Notations.** In this paper, we define $\log(\cdot)$ as the natural logarithm, $\log_2(\cdot)$ as the binary logarithm, $\mathcal{R}^+$ as the set of all positive real numbers, $\mathbb{N}^+$ as the set of all positive integers, $\Delta(\Omega)$ as the probability simplex over $\Omega$, and $[t] = \{1, 2, ..., t\}$. Additionally, we use $a = O(b)$ to indicate the existence of a constant $c'$ such that $c'b \geq a$, and use $\tilde{O}$ to suppress poly-logarithmic factors.

## 3.1. Online information acquisition model

The setting we study was initially explored by Chen et al. (2023), as outlined below.

**Interaction protocol.** The information acquisition model involes three key components: a principal, an agent, and a stochastic environment. The interaction between these entities in each round $t$ proceeds as follows:

1. The principal announces a scoring rule $S_t : \Omega \times \Delta(\Omega) \to \mathcal{R}^+$.

2. The agent selects an action, or "arm", $k_t$ from the $K$-armed set $\mathcal{A}$ based on the scoring rule $S_t$ and incurrs a cost $c_{k_t}$. Note that while $k_t$ is observed by the principal, the cost $c_k$ is private to the agent.

3. The stochastic environment generates a state $\omega_t \in \Omega$ unknown to both the agent and the principal and sends an observation $o_t \in \mathcal{O}$ to the agent based on $p(\omega_t, o_t | k_t)$.

4. The agent submits a reported belief $\hat{\sigma}_t \in \Delta(\Omega)$ to the principal about the hidden state. This belief helps the principal in learning the stochastic environment. The principal then makes a decision $d_t \in \mathcal{D}$ based on knowledge $\{S_t, k_t, \hat{\sigma}_t\}$.

5. The stochastic environment reveals the hidden state $\omega_t$. The principal compensates the agent according to the scoring rule $S_t(\omega_t, \hat{\sigma}_t)$ and gains utility $u(\omega_t, d_t)$.

More details are provided in Section B of the Appendix. We assume the principal selects $S_t$ from the scoring rule set $\mathbb{S}$ where $\|S\|_\infty \le B_S$ for all $S \in \mathbb{S}$, the norm of the utility function $u : \Omega \times \mathcal{D}$ is bounded by $\|u\|_\infty \le B_u$, the arm number, $K \ge 2$, is a finite integer which is known to the principal (Chen et al., 2023), and the observation set $\mathcal{O}$ is also finite with $C_{\mathcal{O}}$ observations.

Note that, in our setting, the principal can observe the action (e.g., research method) selected by the agent (Chen et al., 2023). The practicality of this assumption will be illustrated through the following example.

**Example 1.** *Suppose the agent's action $k_t$ denotes a research method or a type of investigation. The principal can design processes to make the agent's actions observable. Below are practical examples:*

- ***Online Surveys**: The agent deploys a web-based questionnaire to gather information, where each distinct type of web-based questionnaire represents a specific action. The principal can mandate that the questionnaire be hosted on the principal's company's official website or made easily accessible online, enabling the principal to readily observe the process.*

- ***Offline In-Person Surveys**: The agent organizes a group of investigators to conduct face-to-face surveys on the street, where each distinct type of survey represents a specific action. The principal can require that some of these investigators include trusted employees from the principal's company, ensuring that the principal can observe this category of surveys.*

In the rest of the Section 2, we will forget the subscript $t$ and introduce the agent's arm selection and belief reporting policies, along with the decision-making policy of the principal

**Information structure.** The almighty agent is aware of both $\{c_k\}_{k=1}^K$ and the observation generation process $p(\omega, o | k)$. With this knowledge, the agent refines its belief $\sigma \in \Delta(\Omega)$ about the hidden state using the Bayes's rule: $\sigma(\omega) = p(\omega | o, k) = p(\omega, o | k) / p(o | k)$. Here, $\sigma$ is a random measure mapping observations from $\mathcal{O}$ to probabilities in $\Delta(\Omega)$, reflecting the uncertainty in $o$. Let $\Sigma \subset \Delta(\Omega)$ represent the support of $\sigma$, and define $M$ as the cardinality of $\Sigma$. Due to the discrete nature of $\mathcal{A}$ and $\mathcal{O}$, we have $M \le K \times C_{\mathcal{O}}$. We also introduce $q_k(\sigma) \in \Delta(\Sigma)$ as the distribution of $\sigma$ based on the agent's choice of action $k \in \mathcal{A}$. Since $\sigma$ encapsulates all the relevant information about $o$, the observation $o$ can be omitted, and $\{q_k\}_{k=1}^K$ along with $\{c_k\}_{k=1}^K$ are referred to as the information structure.

**Optimal scoring rule.** The information acquisition problem can be framed as a general Stackelberg game. In this setting, let $\eta : \mathbb{S} \times \Delta(\Omega) \times \mathcal{A} \to \mathcal{D}$ represent the principal's decision-making policy, while $\mu : \mathbb{S} \to \mathcal{A}$ and $v : \mathbb{S} \times \Sigma \times \mathcal{A} \to \Delta(\Omega)$ denote the agent's arm selection and belief reporting policies, respectively. For any given scoring rule $S \in \mathbb{S}$, the agent chooses an arm $k = \mu(S)$ and reports $\hat{\sigma} = v(S, \sigma, k)$ to maximize its expected profit, which is expressed as $g(\mu, v; S) := \mathbb{E}\left[S\left(\omega, v\left(S, \sigma, \mu(S)\right)\right) - c_{\mu(S)}\right]$. The expectation here accounts for the randomness in both $\omega$ and $\sigma$ according to the distribution $q_{\mu(S)}$. The principal's objective is to design an optimal scoring rule $S$ and establish a decision policy $\eta$ that maximizes its own expected profit, given that the agent will respond optimally with its best strategies $(\mu^*, v^*)$, i.e., $h\left(\mu^*, v^*; S, \eta\right) := \mathbb{E}\left[u\left(\omega, \eta\left(S, v^*\left(S, \sigma, \mu^*(S)\right), \mu^*(S)\right)\right) - S\left(\omega, v^*\left(S, \sigma, \mu^*(S)\right)\right)\right]$ where the expectation considers the randomness of $\omega$ and $\sigma$ with respect to $q_{\mu^*(S)}$. The optimal leader strategies are referred to as strong Stackelberg equilibria, which can be framed as solutions to a bilevel optimization problem, as introduced by Conitzer (2015); Chen et al. (2023), i.e.,

$$\max_{\eta, S \in \mathbb{S}} h\left(\mu^*, v^*; S, \eta\right),$$
$$\text{s.t. } \left(\mu^*, v^*\right) \in \arg\max_{\mu, v} g\left(\mu, v; S\right). \tag{1}$$

To simplify Eq (1), we will introduce the definition of proper scoring rule (Savage, 1971). Under this rule, the agent is incentivized to report truthfully (i.e., $\hat{\sigma} = \sigma$).

**Definition 1** (Proper scoring rule (Savage, 1971)). *A scoring rule $S \in \mathbb{S}$ is considered proper if, for any given belief $\sigma \in \Delta(\Omega)$ and any reported belief $\hat{\sigma} \in \Delta(\Omega)$, we have $\mathbb{E}_{\omega \sim \sigma}[S(\omega, \hat{\sigma})] \leq \mathbb{E}_{\omega \sim \sigma}[S(\omega, \sigma)]$. If this inequality is true for any $\hat{\sigma} \neq \sigma$, then $S$ is said to be strictly proper.*

Under a proper scoring rule, the concept of reporting a truthful belief is, by definition, the most profitable for the agent. According to the revelation principle (Myerson, 1979), any equilibrium that involves dishonest belief reporting has an equivalent equilibrium involving truthful belief reporting. This implies that the principal's optimal scoring rules can be considered proper without loss of generality (Chen et al., 2023), allowing us to limit the reporting scheme $v$ to the ones that are truthful.

**Lemma 1** (Revelation principal (Myerson, 1979)). *We let $\mathcal{S}$ denote the set of the proper scoring rule with bounded norm $\|S\|_\infty \leq B_S$, there exists a $S^* \in \mathcal{S}$ that is an optimal solution to Eq (1).*

We define $k^* = k(S^*)$ as the optimal response of $S^*$. The proof of Lemma 1 is shown in Section C.1 of Chen et al. (2023). We here further simplify the notations. The optimal decision policy for the principal $\eta^*$ can be simplified to $d^*(\sigma) = \eta^*(S, \sigma, k)$ since $S$ and $k$ add no information to the hidden state given $\sigma$. Similarly, we define $k^*(S) = \mu^*(S)$, $u(\sigma) = \mathbb{E}_{\omega \sim \sigma}[u(\omega, d^*(\sigma))]$ and $S(\sigma) = \mathbb{E}_{\omega \sim \sigma}[S(\omega, \sigma)]$. We can reformulate the optimization program in Eq (1) as follows:

$$\max_{S \in \mathcal{S}} \quad \mathbb{E}_{\sigma \sim q_{k^*(S)}} \big[ u(\sigma) - S(\sigma) \big]$$
$$\text{s.t.} \quad k^*(S) \in \arg\max_{k \in \mathcal{A}} \mathbb{E}_{\sigma \sim q_k} \big[ S(\sigma) - c_k \big]. \quad (2)$$

We denote the agent's profit function as $g(k, S) := \mathbb{E}_{\sigma \sim q_k}[S(\sigma) - c_k]$ and the principal's profit function under the agent's optimal response $k^*(S)$ as $h(S) := \mathbb{E}_{\sigma \sim q_{k^*(S)}}[u(\sigma) - S(\sigma)]$. In Section 4, we will convert Eq (2) into linear programming that only includes learnable parameters.

### 3.2. Learning objectives

In this paper, we consider both the fixed confidence setting and the fixed budget setting.

**Fixed confidence.** In the fixed confidence setting, the algorithm wants to obtain an estimated optimal scoring rule $\hat{S}^*$ that can satisfy the $(\epsilon, \delta)$-condition with a finite sample complexity $\tau$.

**Definition 2** ($(\epsilon, \delta)$-condition). *We define $0 < \epsilon \leq 2(B_S + B_u)$ as the reward gap parameter and $0 < \delta \leq 1$ as the*

*probability parameter. The algorithm should find a estimated optimal scoring rule $\hat{S}^* \in \mathcal{S}$ which satisfies*

$$\mathcal{P}\left( h(S^*) - h(\hat{S}^*) \leq \epsilon \right) \geq 1 - \delta. \quad (3)$$

A smaller sample complexity $\tau$ implies better algorithmic performance.

**Fixed budget.** In the fixed budget setting, the learning algorithm should utilize $T$ interactions to derive an estimated optimal scoring rule $\hat{S}^*$ which satisfies

$$\mathcal{P}\left( h(S^*) - h(\hat{S}^*) \leq \epsilon \right) \geq 1 - \tilde{\delta}. \quad (4)$$

A smaller error probability $\tilde{\delta}$ implies better algorithmic performance.

**Remark 1.** *The best scoring rule identification problem is significantly more challenging than the best arm identification problems in standard multi-armed bandits Even-Dar et al. (2002) and principal-agent bandits Scheid et al. (2024). In the principal-agent bandit setting Scheid et al. (2024), the principal's objective is relatively straightforward: to incentivize the agent to take a good action, as the agent's action directly determines the principal's utility. In contrast, our setting introduces an added layer of complexity due to the presence of an underlying state that is unobservable to the principal but directly impacts the agent's utility. The principal must rely on the agent's report to derive the information about the state. Consequently, maximizing utility in our setting requires addressing two intertwined challenges: it must incentivize the agent to (1) take the optimal action, and (2) truthfully report its belief about the underlying state.*

*Consequently, while the payment rule in Scheid et al. (2024) can be designed solely based on the agent's action, the scoring rule in our setting instead depends on the agent's report. This introduces another layer of difficulty: the scoring rule must not only incentivize the agent to take the optimal action, but also to truthfully report its belief. Hence, identifying the optimal scoring rule in our setting demands a more nuanced and sophisticated analysis.*

## 4. Learning the Optimal Scoring Rule

In this section, we demonstrate how algorithms iteratively learn the optimal scoring rule. In Section 4.1, we will simplify the information acquisition problem into a bandit-like problem, identifying the key parameters to be learned. In Section 4.2, we will delve into the strategy of learning these unknown parameters. In Section 4.3, we will use the learned parameters to establish the linear program UCB-LP$_{k,t}$ and use it to select the estimated best arm.

## 4.1. Simplify the information acquisition problem

We first illustrate how to simplify the information acquisition problem into a bandit-liked problem. Define $\mathcal{V}_k = \{S \in \mathcal{S} \mid g(k,S) \geq g(k',S), \forall k' \in \mathcal{A}\}$ as the region in which the agent takes action $k$ as its best response. The problem Eq (2) can be reformulated as

$$\max_{k \in \mathcal{A}} h(S_k^*)$$
$$\text{s.t.} \quad h(S_k^*) = \sup_{S \in \mathcal{V}_k} \mathbb{E}_{\sigma \sim q_k}[u(\sigma) - S(\sigma)]. \tag{5}$$

where $h(S_k^*)$ represents the principal's maximum profit when the agent's optimal response is $k$, and $S_k^*$ is the optimal solution to the inner problem in Eq (5). If the principal is aware of $\mathcal{V}_k$, $q_k$ and the pairwise cost difference $C(k,k') = c_k - c_{k'}$, the inner problem of Eq (5) can be solved by the following linear programming $\text{LP}_k$

$$\text{LP}_k: \quad h(S_k^*) = \max_{S \in \mathcal{S}} u_k - v_S(k)$$
$$\text{s.t.} \quad v_S(k) - v_S(k') \geq C(k,k'), \quad \forall k \in \mathcal{A}, \tag{6}$$

where $v_S(k) = \langle S(\cdot), q_k(\cdot) \rangle_\Sigma$ denotes the expected payment of scoring rule $S$ if the agent takes arm $k$ and $u_k = \langle u(\cdot), q_k(\cdot) \rangle_\Sigma$ denotes the expected utility if the agent takes arm $k$. The information acquisition problem is simplified to a bandit problem, where $\mathcal{A}$ is the $K$-armed set and $h(S_k^*)$ is the reward of the arm. However, the region $\mathcal{V}_k$, the brief distribution $q_k$ and pairwise cost difference $C(k,k')$ are unknown to the principal. Therefore, the principal needs to learn this information. Recall the definitions of $\mathcal{V}_k$ and $g(k,S)$: $\mathcal{V}_k = \{S \in \mathcal{S} | \langle q_k - q_{k'}, S \rangle_\Sigma \geq C(k,k'), \forall k' \in \mathcal{A}\}$. To identify $\mathcal{V}_k$, we just need to estimate the belief distribution $q_k$ with estimator $\hat{q}_k^t$ and the pairwise cost difference $C(k,k') = c_k - c_{k'}$ with estimator $\hat{C}^t(k,k')$. Note that estimator $\hat{C}^t$ can be updated using the following identity

$$C(k,k') = v_S(k) - v_S(k'), \; \forall S \in \mathcal{V}_k \cap \mathcal{V}_{k'}, \tag{7}$$

where we can replace the $q_k$ in $v_S(k)$ with $\hat{q}_k^t$. Furthermore, we need to find a scoring rule $S$ that belongs to $\mathcal{V}_k \cap \mathcal{V}_{k'}$ to use Eq (7) effectively. To achieve this, we utilize a binary search approach on the convex combination of $S_1$ and $S_2$, where $k^*(S_1) = k$ and $k^*(S_2) = k'$. Besides, we aim to inform the principal about the $K$ arms and ensure they can find a scoring rule $S$ such that $k^*(S) = k$ for binary search. To achieve this, we introduce the action-informed oracle. This oracle provides the principal with $K$ scoring rules $\tilde{S}_1, ..., \tilde{S}_K$ such that $k^*(\tilde{S}_k) = k$. This assumption is also adopted by Chen et al. (2023) where they demonstrated the practical implementation of the action-informed oracle (see Example 3.4 and 3.5 in Chen et al. (2023)). Furthermore, they presented a hard instance in which no learning algorithm can identify the scoring rule that triggers the best arm without the action-informed oracle (Lemma 3.2 in Chen et al. (2023)).

**Assumption 1** (Action-informed oracle Chen et al. (2023)). *We suppose that there is an oracle that can provide the learning algorithm with $K$-scoring rules $\{\tilde{S}_k\}_{k=1}^K$. If the principal announces $\tilde{S}_k$, the agent's best response is arm $k$. Moreover, for the agent's profit $g(k,S)$, we suppose there exists a positive constant $\varepsilon$ that $g(k, \tilde{S}_k) - g(k', \tilde{S}_k) > \varepsilon$, $\forall k' \neq k$.*

Intuitively, equipped with the action-informed oracle, the principal holds the power to induce the agent to select its preferred arm (i.e., announce the corresponding scoring rule). In Section 4.2, we will show that we can utilize $\hat{q}^t$, $\hat{C}^t$, and their corresponding confidence radius to efficiently estimate $q$ and $C$.

## 4.2. Learning belief distributions and pairwise cost differences

Let $\mathcal{N}_k^t = \sum_{s=1}^{t-1} \mathbb{1}\{k_s = s\}$ denote the total count of times the agent selects arm $k$ before round $t$. For all $k \in \mathcal{A}$, we define the empirical estimator of $q_k$ as

$$\hat{q}_k^t(\sigma) = \frac{1}{\mathcal{N}_k^t} \sum_{s=1}^{t-1} \mathbb{1}\{\sigma_s = \sigma, k_s = k\}, \; \forall \sigma \in \Sigma. \tag{8}$$

We define the confidence radius for belief estimator $\hat{q}_k^t$ as

$$\text{Fixed Confidence } I_q^t(k) = \sqrt{\frac{2\log(4K 2^M t^2/\delta)}{\mathcal{N}_k^t}}, \tag{9}$$

$$\text{Fixed Budget } I_q^t(k) = \sqrt{\frac{a}{\mathcal{N}_k^t}}, \tag{10}$$

where the definition of parameter $a$ will be provided in Theorem 2. We define the estimated payment of scoring rule $S$ if the agent responds by taking arm $k$ as $\hat{v}_S^t(k) = \langle S(.), \hat{q}_k^t(.) \rangle_\Sigma$. Besides, we define the estimated utility if the agent takes arm $k$ as $\hat{u}_k^t = \langle u(.), \hat{q}_k^t(.) \rangle_\Sigma$.

The definition of the pairwise cost difference estimator $\hat{C}^t(k,k')$ and its confidence radius $I_c^t(k,k')$ are shown in Section C in the Appendix. By the following Lemma 2, we show that $q_k$ and $C(k,k')$ can be successfully learned with high probability.

**Lemma 2.** *Define event $\mathcal{E} = \{\|\hat{q}_k^t - q_k\|_1 \leq I_q^t(k), \forall t \in [\tau], \forall k \in \mathcal{A}\}$. $\mathcal{E}$ will occur with probability at least $1 - \delta$ when $I_q^t(k)$ is defined in the fixed confidence style Eq (9) and with probability at least $1 - TK 2^M e^{-\frac{1}{2}a}$ (if $0 < TK 2^M e^{-\frac{1}{2}a} \leq 1$) when defined in the fixed budget style Eq (10). Furthermore, under the occurrence of event $\mathcal{E}$, we have:*

$$|\hat{v}_S^t(k) - v_S(k)| \leq B_S I_q^t(k), \; \forall k \in \mathcal{A}, \; S \in \mathcal{S},$$
$$|\hat{u}_k^t - u_k| \leq B_u I_q^t(k), \; \forall k \in \mathcal{A},$$
$$|C(k,k') - \hat{C}^t(k,k')| \leq I_c^t(k,k'), \; \forall (k,k').$$

### 4.3. Solving for the optimal scoring rule via UCB-LP$_{k,t}$

Rewriting the LP$_k$ (6) by replacing $q$ and $C$ with their estimations $\hat{q}^t$, $\hat{C}^t$, and incorporating their confidence radiuses, we obtain

$$\text{UCB-LP}_{k,t}: \quad \hat{h}_k^t = \max_{S \in \mathcal{S}} \hat{u}_k^t + B_u I_q^t(k) - v$$

$$s.t. \quad |v - \hat{v}_S^t(k)| \le B_S I_q^t(k)$$

$$v - \hat{v}_S^t(k') \ge \hat{C}^t(k,k') - \left(I_c^t(k,k') + B_S I_q^t(k')\right), \; \forall k' \neq k.$$

In UCB-LP$_{k,t}$, we employ $\hat{h}_k^t$ to provide a high-probability upper bound on the true value of $h(S_k^*)$, as demonstrated in Lemma 7 in the Appendix. Through the linear programming described above, we can derive the estimated value $\hat{h}_k^t$ and scoring rule $\hat{S}_{k,t}$ for arm $k$ in round $t$. In cases where UCB-LP$_{k,t}$ is infeasible, we set $\hat{h}_k^t = \hat{u}_k^t - \hat{v}_k^t(\tilde{S}_k) + (B_u + B_S) I_q^t(k)$ and $\hat{S}_{k,t} = \tilde{S}_k$.

## 5. Learning Algorithms

### 5.1. Online information acquisition fixed confidence (OIAFC)

Before presenting the learning algorithm, we first define the concepts of the reward gap and problem complexity.

**Definition 3** (Reward gap and problem complexity). *We define the reward gap between the optimal arm and a sub-optimal arm $k$ as $\Delta_k = h(S^*) - h(S_k^*)$. Furthermore, the instance-dependent problem complexity is defined as $H_\Delta = 4(B_S + B_u)^2\left(\epsilon^{-2} + \sum_{k \neq k^*} \Delta_k^{-2}\right)$ and instance-independent problem complexity as $H_\epsilon = 4(B_S + B_u)^2 K/\epsilon^2$.*

The reward gap $\Delta_k$ represents the difference between the maximum profit that the principal can obtain when the agent pulls the optimal arm $k^*$ versus a sub-optimal arm $k$. In the bandit literature, problem complexity is commonly defined as $H_\Delta = \epsilon^{-2} + \sum_{k \neq k^*} \Delta_k^{-2}$ and $H_\epsilon = K\epsilon^{-2}$. This definition arises from the assumption that the rewards of each arm are bounded within $[0, 1]$, leading to $0 \le \Delta_k \le 1$ and $0 \le \epsilon \le 1$. In our context, we have $0 \le \frac{\Delta_k}{2(B_S + B_u)} \le 1$ and $0 \le \frac{\epsilon}{2(B_S + B_u)} \le 1$. This consideration motivates our selection of problem complexity in Definition 3.

We now introduce the OIAFC algorithm.

**OIAFC.** As outlined in lines 2-4 of Algorithm 1, OIAFC begins with an initialization phase, where action-informed oracles of different arms are presented to the agent, feedback is collected, and estimates are updated. The algorithm then enters the exploration phase, where the principal's task is to identify the currently estimated best arm, denoted as $k_t^* = \arg\max_{k \in \mathcal{A}} \hat{h}_k^t$, with $\hat{h}_k^t$ derived from UCB-LP$_{k,t}$. To incentivize the agent to select this arm,

the algorithm employs a conservative scoring rule $S_t = \alpha_{k_t^*}^t \tilde{S}_{k_t^*} + \left(1 - \alpha_{k_t^*}^t\right) \hat{S}_{k_t^*,t}$. After receiving the agent's feedback, the algorithm checks if further exploration is needed. If the agent chooses $k_t \neq k_t^*$, it suggests that the estimate of $\mathcal{V}_{k_t^*}$ is not accurate, prompting a binary search (details can be found in Algorithm 3 in the Appendix) to refine the estimate. Conversely, if $k_t = k_t^*$, this suggests a satisfactory estimate of $\mathcal{V}_{k_t^*}$, and the algorithm proceeds to check the stopping condition. If $2(B_S + B_u)I_q^t(k_t^*) \le \beta_t$ holds, the exploration halts, returning $\hat{S}^* = S_t$ as the estimated optimal scoring rule.

In the following sections, we will illustrate the insight behind the design of Algorithm 1 and establish upper bounds for normal exploration rounds $\sum_{t=1}^\tau \mathbb{1}\{k_t = k_t^*\}$ and forced exploration rounds $\sum_{t=1}^\tau \mathbb{1}\{k_t \neq k_t^*\}$ in Lemma 3 and Lemma 4, respectively. Finally, by utilizing Lemma 5, we demonstrate that OIAFC successfully identifies the estimated best arm that satisfies condition Eq (3). Combining these results, we can prove that OIAFC outputs an estimated best-scoring rule that satisfies Eq (3) with a near-optimal sample complexity (shown in Theorem 1).

**Sampling rule.** In each round, the principal identifies $k_t^* = \arg\max_{k \in \mathcal{A}} \hat{h}_k^t$ as the current estimated best arm and wants to induce the agent to pull the arm $k_t^*$ using the scoring rule $S_t$. However, at times, $\hat{C}^t(k_t^*, k) + I_c^t(k_t^*, k)$ and $\hat{q}_{k_t^*}^t + I_q^t(k_t^*)$ may not be precise enough to estimate $C(k_t^*, k)$ and $q_{k_t^*}$ due to limited observations of arm $k_t^*$. Consequently, the agent might not choose arm $k_t^*$ under scoring rule $\hat{S}_{k_t^*,t}$. This indicates directly setting $S_t = \hat{S}_{k_t^*,t}$ is not sufficient to induce arm $k_t^*$. Hence, we let the principal adopt a conservative scoring rule $S_t = \alpha_k^t \tilde{S}_{k_t^*} + \left(1 - \alpha_k^t\right) \hat{S}_{k_t^*,t}$. Intuitively, since the scoring rule $\tilde{S}_{k_t^*}$ ensures the agent will select arm $k_t^*$, incorporating it with $\hat{S}_{k_t^*,t}$ increases the likelihood that the agent pulls arm $k_t^*$ compared to using $\hat{S}_{k_t^*,t}$ alone, as the combined rule strengthens the agent's preference for the estimated best arm.

**Remark 2.** *Therefore, if $S_t$ contains more ingredients from $\tilde{S}_{k_t^*}$ (meaning $\alpha$ is relatively large), it's more likely to prompt the agent to select $k_t^*$ and requires fewer binary searches. Consider the extreme case that $\alpha_k^t = 1$ for all $t$, then the OIAFC will directly present the action-informed oracle of arm $k$ and the agent must respond with $k_t = k$. However, this might increase the simple regret $h(S^*) - h(S_t)$, leading the algorithm to employ more regular rounds (i.e., $k_t^* = k_t$) to estimate a suitable $\hat{S}^*$ such that it can satisfy the condition in Eq (3). Therefore, selecting suitable $\{\alpha_k^t\}_{k=1}^K$ and $\beta_t$ to balance the trade-off between normal exploration (i.e., $\tau_1 = \sum_{t=1}^\tau \mathbb{1}\{k_t = k_t^*\}$) and forced exploration (i.e., $\tau_2 = \sum_{t=1}^\tau \mathbb{1}\{k_t \neq k_t^*\}$, note that rounds $k_t \neq k_t^*$ include the rounds in the binary search) is one of the key challenges addressed in this paper.*

---

**Algorithm 1** Online Information Acquisition Fixed Confidence (OIAFC)

---
1: **Inputs:** Arm set $\mathcal{A}$, action-informed oracle $\{\tilde{S}_k\}_{k=1}^K$
2: **for** $t = 1 : K$ **do**
3:      Principal announces $S_t = \tilde{S}_t$ and updates the data based on its observation             ▷ Initialization
4: **end for**
5: **while** $t < \infty$ **do**
6:      Calculate $\hat{q}_k^t$, $\hat{C}^t(k, k')$, $I_q^t(k)$ in Eq (9), $I_c^t(k, k')$ for all $k \in \mathcal{A}$ and pair $(k, k')$
7:      **for** $k = 1 : K$ **do**
8:          Solve UCB-LP$_{k,t}$ and derive $\hat{h}_t^k$ and $\hat{S}_{k,t}$
9:      **end for**
10:     Set $k_t^* = \arg\max_{k \in \mathcal{A}} \hat{h}_k^t$ and $S_t = \alpha_{k_t^*}^t \tilde{S}_{k_t^*} + (1 - \alpha_{k_t^*}^t) \hat{S}_{k_t^*, t}$           ▷ Sampling rule
11:     Principal announces $S_t$ and observes the responding arm of the agent $k_t$
12:     **if** $k_t \neq k_t^*$ **then**
13:         Conduct binary search BS$(S_t, \tilde{S}_{k_t^*}, k_t^*, t)$                 ▷ Forced exploration
14:     **else**
15:         **if** $2(B_S + B_u) I_q^t(k^*) \leq \beta_t$ **then**
16:            Output $\hat{S}^* = S_t$ as the estimated best scoring rule and break       ▷ Decision rule
17:         **end if**
18:     **end if**
19:     Set $t = t + 1$ and begin a new round
20: **end while**

---

Define $\mathcal{L}_k^t = \sum_{s=1}^t \mathbb{1}\{k_s = k_s^*, k_s = k\}$ as the number that $k$ appears in a normal exploration till round $t$. With the following Lemma 3, we can derive a instance-dependent upper bound of $\tau_1$.

**Lemma 3.** *If we select* $\alpha_k^t = \min\left(\sqrt{\frac{M}{\mathcal{L}_k^t}}, 1\right)$, $\beta_t = \frac{\epsilon - 2\alpha_{k_t^*}^t(B_S + B_u)}{1 - \alpha_{k_t^*}^t}$ *when* $\alpha_{k_t^*}^t < 1$ *and* $\beta_t = 0$ *when* $\alpha_{k_t^*}^t = 1$, *under event* $\mathcal{E}$, *we have* $\tau_1 = \tilde{O}(M H_\Delta)$.

The result described above is related to the definition of $\alpha_k^t$, the stopping rule outlined in line 15 of Algorithm 1, as well as the structure of the fixed confidence style confidence radius in Eq (9).

**Forced exploration.** If the agent does not choose $k_t = k_t^*$ after the principal announces $S_t$, it indicates that our estimate of $\mathcal{V}_{k_t^*}$ (the second constraint of UCB-LP$_{k_t^*, t}$) is insufficient. To ensure the agent selects $k_t^*$ in future rounds, further exploration is required to improve the estimate of $\mathcal{V}_{k_t^*}$. Specifically, using the structure $S_t = \alpha_{k_t^*}^t \tilde{S}_{k_t^*} + (1 - \alpha_{k_t^*}^t) \hat{S}_{k_t^*, t}$, we update the boundary of $\mathcal{V}_{k_t^*}$ between $S_t$ and $\tilde{S}_{k_t^*}$. This is done by performing a binary search along the region connecting $S_t$ and $\tilde{S}_{k_t^*}$ to determine where the agent's preferred arm shifts from $k_t^*$ to another arm. This binary search efficiently refines the estimate of $\mathcal{V}_{k_t^*}$ in logarithmic time. More details on the binary search algorithm can be found in Algorithm 3 in the Appendix.

In the following Lemma 4, we provide an instance-dependent upper bound of $\tau_2$.

**Lemma 4.** *If we select* $\alpha_k^t$ *and* $\beta_t$ *as Lemma 3, under event* $\mathcal{E}$, *we have* $\tau_2 = \tilde{O}(\varepsilon^{-2} B_S^2 M H_\Delta)$.

**Remark 3.** *One of the key technical contributions of this paper is the introduction of new trade-off parameters* $\{\alpha_k^t\}_{k=1}^K$ *and breaking parameter* $\beta_t$, *which are used to derive the instance-dependent sample complexity upper bound. Chen et al. (2023) set* $\alpha_k^t = \min\left(\frac{K}{t^{1/3}}, 1\right)$ *for all* $k \in \mathcal{A}$, *which limits them to a sub-optimal, instance-independent sample complexity upper bound. In contrast, our parameters ensure that* $\mathcal{L}_k^\tau = \tilde{O}(M 2(B_S + B_u)^2 \Delta_k^{-2})$ *for all* $k \neq k^*$ *(see the proof of Lemma 4 in the Appendix), allowing us to establish an instance-dependent upper bound for* $\tau_1$. *Moreover, these parameters link* $\tau_1$ *and* $\tau_2$, *demonstrating that the number of binary searches can be bounded by* $\tilde{O}(\tau_1 B_S^2 \varepsilon^{-2})$ *(see Section E.2 in the Appendix). Ultimately, this enables us to derive an instance-dependent upper bound for* $\tau = \tau_1 + \tau_2$.

The subsequent Lemma 5 demonstrates that the returned estimated best scoring rule $\hat{S}^*$ can meet the $(\epsilon, \delta)$-condition in Eq (3).

**Lemma 5.** *If we select* $\alpha_k^t$ *and* $\beta_t$ *as Lemma 3, the returned estimated optimal scoring rule of OIAFC (i.e.,* $\hat{S}^*$) *can satisfy the* $(\epsilon, \delta)$-condition in Eq (3).

Together with Lemma 3-5, we can provide the instance-dependent learning performance of the OIAFC.

**Theorem 1.** *We set* $\alpha_k^t = \min\left(\sqrt{\frac{M}{\mathcal{L}_k^t}}, 1\right)$, $\beta_t = \frac{\epsilon - 2\alpha_{k_t^*}^t(B_S + B_u)}{1 - \alpha_{k_t^*}^t}$ *when* $\alpha_{k_t^*}^t < 1$ *and* $\beta_t = 0$ *when* $\alpha_{k_t^*}^t = 1$. *Under such circumstance, OIAFC can output an estimated*

best scoring rule which satisfies the $(\epsilon, \delta)$-condition in Eq (3). The sample complexity of the algorithm can be upper bounded by $\tau = \tilde{O}\left(\varepsilon^{-2} B_S^2 M H_\Delta\right)$ with probability at least $1 - \delta$.

**Remark 4** (Theorem 1 provides a sample complexity upper bound that is near-optimal). *The presence of the $M$ term in our sample complexity upper bound arises because $q_k$ is a probability distribution supported on a finite set of cardinality $M$. To bound $q_k$, we rely on a concentration inequality (Lemma 6 in the Appendix). Consider a simple setting where the belief set is $\{\sigma_i\}_{i=1}^{2K}$ and the agent returns belief $\sigma_{2k-1}$ or $\sigma_{2k}$ to the principal only when it pulls arm $k \in \mathcal{A}$ (this knowledge is known by the principal). Let $q_k$ represents the distribution of $\sigma_{2k-1}$ and $\sigma_{2k}$ based on the agent's action $k$. By replacing the previous belief estimator with $\hat{q}_k^t(\sigma) = \frac{1}{\mathcal{N}_k^t} \sum_{s=1}^{t-1} \mathbf{1}\{\sigma_s = \sigma, k_s = k\}$, $\sigma \in \{\sigma_{2k-1}, \sigma_{2k}\}$, and defining the confidence radius as $I_q^t(k) = \sqrt{\frac{2\log(16Kt^2/\delta)}{\mathcal{N}_k^t}}$, we can upper bound the sample complexity by $\tilde{O}\left(\varepsilon^{-2} B_S^2 H_\Delta\right)$. This result aligns with sample complexity upper bounds of common best arm identification algorithms in the MAB literature (i.e., $\tilde{O}\left(H_\Delta\right)$), e.g., Even-Dar et al. (2006); Gabillon et al. (2012); Jamieson et al. (2013).*

**Remark 5** (Origin of the $1/\epsilon^2$ term in the sample complexity bound). *To find the near-optimal scoring rule, we must identify both the best arm $k^*$ and the set of scoring rules that can trigger the best arm $\mathcal{V}_{k^*}$. Intuitively, the dependency on $1/\Delta_i^2$ arises from identifying the best arm, whereas the dependency on $1/\epsilon^2$ emerges because, even after identifying the best arm, additional samples are required to estimate the set $\mathcal{V}_{k^*}$ at an $\epsilon$-accurate level. Only then can we determine a scoring rule that meets the required accuracy.*

**Remark 6** (Advantages of our algorithm over existing approaches). *We now provide intuition for why our algorithm can achieve near-optimal instance-dependent sample complexity, while Chen et al. (2023); Cacciamani et al. (2023) fails to do so. The primary reason for the suboptimal sample complexity in Chen et al. (2023), aside from choosing an inappropriate parameter, is the adoption of a suboptimal termination criterion. Specifically, they employ the breaking rule originally introduced by Jin et al. (2018), which terminates the algorithm after $\tilde{O}(\varepsilon^{-6} K^6 C_{\mathcal{O}}^3)$ iterations and selects the scoring rule identified in the final round as the best estimate. This termination strategy is inherently inefficient because it intuitively demands a large number of samples to ensure the accuracy of the identified scoring rule. In contrast, our proposed breaking rule (line 15 in Algorithm 1) and corresponding decision rule (line 16 in Algorithm 1) enable a more efficient evaluation of whether a scoring rule satisfies the necessary conditions. Consequently, our approach achieves significantly improved sample complexity.*

*For Cacciamani et al. (2023), we believe their explore-then-commit (ETC) algorithm can achieve a sample complexity bound of $K/\epsilon^2$ by appropriately calibrating the length of the exploration phase with respect to $\epsilon$. However, obtaining an instance-dependent sample complexity bound from ETC-based methods is challenging due to the necessity of setting the exploration phase length in advance, without prior knowledge of the actual instance-specific reward gaps or problem complexity.*

*In summary, our algorithm achieves near-optimal instance-dependent sample complexity due to: (1) the selection of appropriate parameters, (2) the design of suitable breaking and decision rules, and (3) the adoption of an effective algorithm structure (UCB).*

By selecting suitable $\alpha_k^t$ and $\beta_t$, we can also derive an instance-independent sample complexity upper bound. This is tailored to the circumstances that $\Delta_k$ is relatively small and $\epsilon$ is relatively large.

**Corollary 1.** *We set $\alpha_k^t = \frac{\epsilon}{4(B_S + B_u)}$ for all $k \in \mathcal{A}$ and $t \in [\tau]$ and $\beta_t = \frac{\epsilon - 2\alpha_{k_t^*}^t (B_S + B_u)}{1 - \alpha_{k_t^*}^t}$. OIAFC can output an estimated best scoring rule which satisfies the $(\epsilon, \delta)$-condition in Eq (3). The sample complexity of the algorithm can be upper bounded by $\tau = \tilde{O}\left(\varepsilon^{-2} B_S^2 M H_\epsilon\right)$ with probability at least $1 - \delta$.*

The proof of Corollary 1 follows a similar approach to the proof of Theorem 1, i.e., we separately upper bound $\tau_1$ and $\tau_2$, and demonstrate that the estimated best arm satisfies Eq (3). In Chen et al. (2023), the authors show that their algorithm can find the $(\epsilon, \delta)$-optimal scoring rule with at most $\tau = \tilde{O}\left(8(B_S + B_u)^3 C_{\mathcal{O}}^3 K^6 \epsilon^{-3} \varepsilon^{-6}\right)$ samples (see their Corollary 4.4), which is significantly larger than our sample complexity upper bound $\tilde{O}\left(M H_\epsilon \varepsilon^{-2}\right)$ (note that $M \leq K \times C_{\mathcal{O}}$) in Corollary 1. Our results also align with the efficient instance-independent bound in MAB, i.e., $\tau = \tilde{O}\left(K \epsilon^{-2}\right)$ (Even-Dar et al., 2002; Kalyanakrishnan et al., 2012).

### 5.2. Online information acquisition fixed budget (OIAFB)

**OIAFB.** The OIAFB algorithm shares the same sampling rule and forced exploration strategy as OIAFC. However, it diverges in its stopping and decision rules. OIAFB stops once it exhausts the allocated budget $T$, returning the estimated best scoring rule as $\hat{S}^* = S_{t^*}$, where $t^* = \arg\min_{t \in \mathcal{T}} 2(B_S + B_u) I_q^t(k_t^*)$ and $\mathcal{T}$ denotes the set of all $t$ satisfying $k_t = k_t^*$. Another difference is that OIAFB utilizes confidence radius in Eq (10) to design UCB-LP$_{k,t}$. The pseudocode for OIAFB is provided in Algorithm 2 in the Appendix, and its theoretical performance is detailed in Theorem 2 below.

**Theorem 2.** *If we run OIAFB with $\alpha_k^t = \frac{\epsilon}{4(B_S + B_u)}$ for all $t \in [T]$ and $k \in \mathcal{A}$, and $T$ satisfies*

$$2\log\left(TK2^M\right) < \frac{T - K}{144 H_\epsilon \max\left(B_S^2 \varepsilon^{-2}, 1\right) \log_2(T)}.$$

*Choose parameter $a$ such that*

$$2\log(TK2^M) \le a < \frac{T - K}{144 H_\epsilon \max\left(B_S^2 \varepsilon^{-2}, 1\right) \log_2(T)},$$

*then OIAFB can output a $(\epsilon, \tilde{\delta})$-estimated optimal scoring rule with $\tilde{\delta} \le TK2^M e^{-\frac{1}{2}a} \le 1$.*

Theorem 2 can directly lead to the following corollary:

**Corollary 2.** *If we set $\delta \in (0, 1]$,*

$$\begin{aligned}
T = &288 H_\epsilon \max\left(B_S^2 \varepsilon^{-2}, 1\right) \log_2(\Gamma) \\
&\times \left(M + \log\left(\frac{\Gamma K}{\delta}\right)\right) + K,
\end{aligned}$$

*where*

$$\Gamma = \left(1152 H_\epsilon \max(B_S^2 \varepsilon^{-2}, 1)\left(M + (K/\delta)^{1/2}\right) + K\right)^2,$$

*$\alpha_k^t = \frac{\epsilon}{4(B_S + B_u)}$ for all $t \in [T]$ and $k \in \mathcal{A}$, and $a = 2\log\left(TK2^M/\delta\right)$. Then OIAFB can output a $(\epsilon, \tilde{\delta})$-estimated optimal scoring rule with $\tilde{\delta} \le \delta$.*

Corollary 2 demonstrates that, given a budget $T = \tilde{O}(\varepsilon^{-2} B_S^2 M H_\epsilon)$, the OIAFB algorithm can identify an estimated best arm that satisfies the condition in Eq 3, aligning with the results stated in Corollary 1. More generally, the analysis presented in this paper suggests that the performance of the OIAFC and OIAFB algorithms for scoring rule identification is characterized by the same complexity notion (instance-independent) across both fixed confidence and fixed budget settings.

## 6. Conclusion

In this paper, we addressed the Best Scoring Rule Identification problem, building on the framework and results from Chen et al. (2023). We proposed two algorithms: OIAFC for the fixed-confidence setting and OIAFB for the fixed-budget setting. Our analysis shows that OIAFC can identify the $(\epsilon, \delta)$-estimated optimal scoring rule, achieving an instance-dependent upper bound of $\tilde{O}(\varepsilon^{-2} B_S^2 M H_\Delta)$ and an instance-independent upper bound of $\tilde{O}(\varepsilon^{-2} B_S^2 M H_\epsilon)$. The OIAFB algorithm demonstrates similar efficiency in the fixed-budget setting. A promising avenue for future work would be to extend the Best Scoring Rule Identification framework to more complex multi-agent environments, as explored in Cacciamani et al. (2023).

## Impact Statement

This paper presents work whose goal is to advance the field of Machine Learning. There are many potential societal consequences of our work, none which we feel must be specifically highlighted here.

## Acknowledgements

Huazheng Wang is supported by National Science Foundation under grant IIS-2403401.

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

# A. Notations

| | |
|---|---|
| $\mathcal{A}$ | Arm set |
| $K$ | Number of arms |
| $t$ | Time index |
| $\tau$ | Terminated round in the fixed confidence setting |
| $T$ | Budget in the fixed budget setting |
| $\omega_t$ | State |
| $\sigma_t$ | True belief of the agent |
| $\hat{\sigma}_t$ | Reported belief of the agent |
| $S_t$ | Scoring rule |
| $k_t$ | Agent's action |
| $k_t^*$ | Principal's desired arm |
| $h(S)$ | Principal's profit function under the scoring rule $S$ |
| $g(S,k)$ | Agent's profit function under the scoring rule $S$ and arm $k$ |
| $\mathcal{V}_k$ | Set of the scoring rule that can trigger the agent to respond by taking arm $k$ |
| $k^*$ | Best response of $S^*$ |
| $S^*$ | Best scoring rule |
| $\hat{S}^*$ | Estimated best scoring rule |
| $q_k$ | Belief distribution of arm $k$ |
| $c_k$ | Cost of arm $k$ |
| $C(k,k')$ | Cost difference of pair $(k,k')$ |
| $\hat{q}_k^t$ | Estimator of $q_k$ |
| $\hat{C}^t(k,k')$ | Estimator of $C(k,k')$ |
| $I_q^t(k)$ | Confidence radius of $\hat{q}_k^t$ |
| $I_c^t(k,k')$ | Confidence radius of $\hat{C}^t(k,k')$ |
| $\delta$ | Probability parameter of the fixed-confidence setting |
| $\tilde{\delta}$ | Probability parameter of the fixed-budget setting |
| $\epsilon$ | Utility gap $(h(S^*) - h(\hat{S}^*))$ parameter of the fixed-confidence pure exploration problem |
| $\varepsilon$ | Utility gap parameter of the action-informed oracle |
| $\hat{S}_{k,t}$ | Scoring rule solving by UCB-LP$_{k,t}$ |
| $\{\tilde{S}_k\}_{k=1}^K$ | Action-informed oracle |
| $\hat{h}_k^t$ | Estimator of $h(S_k^*)$ |
| $\beta_t$ | Breaking threshold in round $t$ |
| $\alpha_k^t$ | Trade-off parameter of arm $k$ |
| $v_S(k)$ | Expected payment of the scoring rule $S$ if the agent responds by taking arm $k$ |
| $\hat{v}_S^t(k)$ | Estimated payment of the scoring rule $S$ in round $t$ if the agent responds by taking arm $k$ |
| $u_k$ | Expected utility of the principal under arm $k$ |
| $\hat{u}_k^t$ | Estimated utility of the principal under arm $k$ |
| $\mathcal{N}_k^t$ | Number of rounds that arm $k$ is observed till round $t$ |
| $B_S$ | Upper bound of $\|S\|_\infty$ |
| $B_u$ | Upper bound of $\|u\|_\infty$ |
| $H_\Delta$ | Instance-dependent problem complexity |
| $H_\epsilon$ | Instance-independent problem complexity |

## B. Details About The Iteration Processes and The Information Structure

To formulate the problem of optimally acquiring information within a principal-agent framework, we model the system as a stochastic environment, a principal and an agent. In each round $t$, the environment will generate a hidden state $\omega_t \in \Omega$, which is depended on the agent action, and will influence the principal's utility. However, this hidden state remains unknown to both the principal and the agent until the end of the round.

The principal's goal is to elicit the agent's belief about this hidden state. To achieve this, the principal first offers a scoring rule $S_t$ that provides a payment based on the quality of the agent's reported belief. This reward incentivizes the agent to provide a more precise estimation of the hidden state. In response to this incentive, the agent incurs a cost $c_{k_t}$ to take an action (arm) $k_t$ from its $K$-armed set $\mathcal{A}$, gather an observation $o_t$ regarding the hidden state from the stochastic environment, and subsequently generate a belief report $\hat{\sigma}_t$ for the principal.

This belief report, in turn, aids the principal to make a decision $d_t$. At the end of each round, the hidden state is revealed to both agent and principal, enabling the principal to compute the corresponding utility. Based on this, the agent is payed according to the scoring rule initially provided by the principal.

The following table displays the information structure of the online information acquisition problem.

| Public Info | Agent's Info | Principal's Info | Delayed Info |
|---|---|---|---|
| $S_t, k_t, \hat{\sigma}_t$ | $\{c_k\}_{k=1}^K, \{q_k\}_{k=1}^K, o_t, \sigma_t$ | $u$ | $\omega_t$ |

*Table 1.* This table outlines the different types of information in the model. `Public Info` represents information that is accessible to both the principal and the agent at each round. `Agent's Info` consists of information known only to the agent, while `Principal's Info` pertains to information exclusive to the principal. Lastly, `Delayed Info` refers to the hidden state $\omega_t$, which remains unknown throughout the round but becomes revealed at the end of round $t$.

## C. Learning The Pairwise Cost Differences

In this section, we introduce how to utilize estimator $\hat{C}^t(k, k')$ to learn the pairwise cost difference $C(k, k')$. We define for each pair $(k, k')$,

$$C_+^t(k, k') = \min_{s < t : k_s = k} \left( \hat{v}_{S_s}^t(k) - \hat{v}_{S_s}^t(k') \right) + B_S \left( I_q^t(k) + I_q^t(k') \right),$$

$$C_-^t(k, k') = \max_{s < t : k_s = k'} \left( \hat{v}_{S_s}^t(k) - \hat{v}_{S_s}^t(k') \right) - B_S \left( I_q^t(k) + I_q^t(k') \right).$$

We also define

$$\theta^t(k, k') = \frac{C_+^t(k, k') + C_-^t(k, k')}{2}, \quad \phi^t(k, k') = \left| \frac{C_+^t(k, k') - C_-^t(k, k')}{2} \right|. \tag{11}$$

Consider a graph $\mathcal{G} = (B, E)$, where the nodes in $B$ represent the $K$ arms, and the edges in $E$ represent the pairwise cost differences $C(k, k') = c_k - c_{k'}$. Additionally, introduce $\phi^t(k, k')$ as the "length" assigned to the edge $C(k, k')$, indicating the uncertainty when using $\theta^t(k, k')$ to estimate the cost difference. To estimate $C(k, k')$ with minimal error, we aim to find the shortest path between arms $k$ and $k'$ on the graph $\mathcal{G}$. This means we're looking for the shortest path between the arms to minimize the errors in estimating their cost difference. Based on this idea, we can estimate the pairwise cost difference by

$$l_{k,k'} = \text{shortest path}(\mathcal{G}, k, k'),$$

$$\hat{C}^t(k, k') = \sum_{(i,j) \in l_{k,k'}} \theta^t(i, j),$$

$$I_c^t(k, k') = \sum_{(i,j) \in l_{k,k'}} \phi^t(i, j),$$

where $l_{k,k'}$ denotes the shortest path between arm $k$ and $k'$ in graph $\mathcal{G}$, $I_c^t(k, k')$ is the confidence radius of the pairwise cost estimator $\hat{C}^t(k, k')$. Besides, it is evident that $\hat{C}^t(k, k') = -\hat{C}^t(k, k')$ due to $l_{k,k'} = l_{k',k}$ and $\theta^t(k, k') = -\theta^t(k', k)$.

# D. Omitted Algorithms

## D.1. OIAFB

---

**Algorithm 2** Online Information Acquisition-Fixed Budget (OIAFB)

---

1: **Inputs:** Arm set $\mathcal{A}$, action-informed oracle $\{\tilde{S}_k\}_{k=1}^K$, and budget $T$
2: **for** $t = 1 : K$ **do**
3:      Principal announces $S_t = \tilde{S}_t$ and updates the data based on its observation             ▷ Initialization
4: **end for**
5: **while** $t \leq T$ **do**
6:      Calculate $\hat{q}_k^t$, $\hat{C}^t(k, k')$, $I_q^t(k)$ in Eq (10), $I_c^t(k, k')$ for all $k \in \mathcal{A}$ and pair $(k, k')$
7:      **for** $k = 1 : K$ **do**
8:          Solve UCB-LP$_{k,t}$ and derive $\hat{h}_t^k$ and $\hat{S}_{k,t}$
9:      **end for**
10:      Set $k_t^* = \arg\max_{k \in \mathcal{A}} \hat{h}_k^t$ and $S_t = \alpha_k^t \tilde{S}_{k_t^*} + (1 - \alpha_k^t)\hat{S}_{k_t^*, t}$            ▷ Sampling rule
11:      Principal announces $S_t$ and observes the responding arm of the agent $k_t$
12:      **if** $k_t = k_t^*$ **then**
13:          Conduct binary search BS$(S_t, \tilde{S}_{k_t^*}, k_t^*, t)$            ▷ Forced exploration
14:      **end if**
15:      Set $t = t + 1$ and begin a new round
16: **end while**
17: Define set $\mathcal{T}$ as the set of all $t$ that satisfies $k_t = k_t^*$
18: Set $t^* = \arg\min_{t \in \mathcal{T}} 2(B_S + B_u)I_q^t(k_t^*)$
19: Output $\hat{S}^* = S_{t^*}$ as the estimated best scoring rule            ▷ Decision rule

---

## D.2. Binary search

---

**Algorithm 3** Binary Search BS$(S_0, S_1, k^*(S_0), k^*(S_1), t, \{I_q^t(k)\}_{k=1}^K)$

---

1: **Inputs:** $S_0, S_1, k^*(S_0), k^*(S_1), t, \{I_q^t(k)\}_{k=1}^K$
2: Initiate $\lambda_{\min} = 0$, $\lambda_{\max} = 1$, and $t_i = t$
3: **while** $\lambda_{\max} - \lambda_{\min} \geq \min(I_q^{t_i}(k^*(S_0)), I_q^{t_i}(k^*(S_1)))$ **do**
4:      Start a new round $t = t + 1$
5:      Set $\lambda = (\lambda_{\min} + \lambda_{\max})/2$ as the middle point
6:      The principal announces $S_t = (1 - \lambda)S_0 + \lambda S_1$ and obtain the agent's response $k_t$
7:      **if** $k_t = k^*(S_1)$ **then**
8:          Set $\lambda_{\max} = \lambda$
9:      **else**
10:          Set $\lambda_{\min} = \lambda$
11:      **end if**
12: **end while**
13: **Return:** $t$

---

**Binary search** In binary search, the algorithm continues searching for the initial point of transition where the agent switches from responding to arm $k^*(S_1)$ to another arm within the region $(S_0, S_1)$. $\lambda_{\max}$ and $\lambda_{\min}$ define the interval containing the switching point. After each round, the binary search assigns $\lambda = (\lambda_{\max} + \lambda_{\min})/2$ to either $\lambda_{\min}$ or $\lambda_{\max}$, effectively halving the candidate interval. The binary search stops once the searching error $\lambda_{\max} - \lambda_{\min}$ becomes smaller than the minimum confidence radius between $k^*(S_1)$ and $k^*(S_2)$. It's important to note that the depth of the binary search, denoted as $m$, is less than $\log_2 \left( \frac{1}{\min(I_q^t(k^*(S_0)), I_q^t(k^*(S_1)))} \right)$ (also less than $\log_2(t)$ by the definition of the confidence interval). Therefore, the binary search ensures sufficient accuracy within a logarithmic time.

**New notations** We hereby introduce new notations to describe more details of the binary search. We define a binary search operation as BS$(S_0, S_1, \tilde{\lambda}_{\max}, \tilde{\lambda}_{\min}, k_0, k_1, t_0, t_1)$, wherein $S_0$ and $S_1$ denote the input scoring rules. Notation

$(\tilde{\lambda}_{\max}, \tilde{\lambda}_{\min})$ represents the $(\lambda_{\max}, \lambda_{\min})$ at the end of this binary search. Additionally, $k_0$ and $k_1$ are determined as $k^*\big((1 - \tilde{\lambda}_{\min})S_0 + \tilde{\lambda}_{\min}S_1\big)$ and $k^*\big((1 - \tilde{\lambda}_{\max})S_0 + \tilde{\lambda}_{\max}S_1\big)$, respectively. Lastly, $t_0$ and $t_1$ signify the initiation and culmination rounds of the binary search process. It's important to note that while $k_1$ represents the optimal response under $S_1$, $k_0$ may not necessarily be the optimal response under $S_0$.

## E. Proof of Lemma 2

To prove Lemma 2, we need to employ the concentration result of Mardia et al. (2018).

**Lemma 6** (Mardia et al. (2018), Lemma 1, Concentration for empirical distribution). *Let $X$ denote a probability distribution that is supported in a finite set with a cardinality of at most $M$. Let $X_1, X_2, \ldots, X_n$ be i.i.d. random variables drawn from $X$. The sample mean is denoted by $\bar{X} = \frac{1}{n}\sum_{s=1}^{n} X_s$. For any sample size $n \in \mathbb{N}^+$ and $0 \leq \delta \leq 1$, the following holds:*

$$\mathcal{P}\left(\|X - \bar{X}\|_1 \geq \sqrt{\frac{2\log(2^M/\delta)}{n}}\right) \leq \delta. \tag{12}$$

*Proof of Lemma 2.* Recalling the event $\mathcal{E} := \big\{\|\hat{q}_k^s - q_k\|_1 \leq I_q^s(k), \ \forall s \in [t], \ \forall k \in \mathcal{A}\big\}$, we define $\mathcal{E}^c$ as the complement of $\mathcal{E}$. By applying the union bound, we obtain:

$$\mathcal{P}(\mathcal{E}^c) \leq \sum_{k=1}^{K}\sum_{s=1}^{t} \mathcal{P}\Big(\|\hat{q}_k^s - q_k\|_1 > I_q^s(k)\Big). \tag{13}$$

Based on Lemma 6 and the definition of the fixed-confidence confidence radius $I_q^s(k)$ in Eq (9), it follows that $\mathcal{P}\Big(\|\hat{q}_k^s - q_k\|_1 > I_q^s(k)\Big) \leq \delta/(4Ks^2), \ \forall s \in [t], \ k \in \mathcal{A}$. Substituting this result into Eq (13), we obtain:

$$\mathcal{P}(\mathcal{E}^c) \leq \sum_{k=1}^{K}\sum_{s=1}^{t} \frac{\delta}{4Ks^2} \leq \delta. $$

It implies $\mathcal{P}(\mathcal{E}) \geq 1 - \delta$.

Employing Lemma 6 and following a procedure analogous to the previous one, it is straightforward to demonstrate that $\mathcal{P}(\mathcal{E}) \geq 1 - tK2^M e^{-\frac{1}{2}a}$ if $I_q^s(k)$ is defined in the fixed-budget style, as in Eq (10).

Consider the case when $\mathcal{E}$ occurs. Based on $\|S\|_\infty \leq B_S$ and $\|u\|_\infty \leq B_u$, it follows that

$$\begin{aligned} |\hat{v}_S^t(k) - v_S(k)| = \big|\langle \hat{q}_k^t - q_k, S\rangle_\Sigma\big| &\leq B_S I_q^t(k), \quad \forall S \in \mathcal{S}, \ k \in \mathcal{A}, \\ \big|\hat{u}_k^t - u_k\big| = \big|\langle \hat{q}_k^t - q_k, u\rangle_\Sigma\big| &\leq B_u I_q^t(k), \quad \forall k \in \mathcal{A}. \end{aligned} \tag{14}$$

Besides, based on the definition of $C_+^t$ and $C_-^t$, we have

$$\begin{aligned} C_+^t(k, k') &\geq \min_{s < t: k_s = k} v_{S_s}(k) - v_{S_s}(k') \geq C(k, k'), \\ C_-^t(k, k') &\leq \max_{s < t: k_s = k'} v_{S_s}(k) - v_{S_s}(k') \leq C(k, k'). \end{aligned} \tag{15}$$

Utilize Eq (15), we have

$$
\begin{aligned}
\left| \hat{C}^t(k,k') - C(k,k') \right| &= \left| \sum_{(i,j)\in l_{k,k'}} \left( \frac{C_+^t(i,j) + C_-^t(i,j)}{2} - C(i,j) \right) \right| \\
&= \left| \sum_{(i,j)\in l_{k,k'}} \left( \frac{1}{2}\Big(C_+^t(i,j) - C(i,j)\Big) + \frac{1}{2}\Big(C_-^t(i,j) - C(i,j)\Big) \right) \right| \\
&\le \frac{1}{2}\left| \sum_{(i,j)\in l_{k,k'}} \Big(C_+^t(i,j) - C(i,j)\Big) \right| + \frac{1}{2}\left| \sum_{(i,j)\in l_{k,k'}} \Big(C_-^t(i,j) - C(i,j)\Big) \right| \\
&\le \frac{1}{2}\sum_{(i,j)\in l_{k,k'}} \Big( |C_+^t(i,j) - C(i,j)| + |C_-^t(i,j) - C(i,j)| \Big) \\
&= \frac{1}{2}\sum_{(i,j)\in l_{k,k'}} |C_+^t(i,j) - C_-^t(i,j)| = I_c^t(k,k'),
\end{aligned}
$$

where the first and second inequalities follow from the triangle inequality, the third equality follows from Eq (15), and the last equality follows from the definition of $I_c^t(k,k')$. This completes the proof of Lemma 2. $\qquad\square$

## F. Proof of Lemma 3-5 and Theorem 1

### F.1. Proof of Lemma 3

*Proof of Lemma 3.* We will break this proof into three steps. In the first step, we upper bound $\sum_{k\neq k^*} \mathcal{L}_k^\tau$. In the second step, we upper bound $\mathcal{L}_{k^*}^\tau$. Finally, in the third step, we upper bound $\tau_1 = \sum_{k=1}^K \mathcal{L}_k^\tau$.

**First step:** We want to prove

$$
\mathcal{L}_k^\tau \le \frac{\left( (2\sqrt{2}+8)(B_S+B_u)\sqrt{\log(4K2^M\tau^2/\delta)} \right)^2}{\Delta_k^2}, \quad \forall k \neq k^*. \tag{16}
$$

Define arm $k \neq k^*$ and let $t_k$ be the final round in which arm $k$ is observed, with $k = k_{t_k}^* = k_{t_k}$. Then, based on Lemma 2 and Lemma 8, when $\alpha_k^{t_k} < 1$, we can derive

$$
\frac{1}{1-\alpha_k^{t_k}}\Big(h(S_k^*) - \alpha_k^{t_k} h(\tilde{S}_k)\Big) + 2(B_S+B_u)I_q^{t_k}(k) \ge \hat{h}_{t_k}^k \ge \hat{h}_{t_k}^{k^*} \ge h(S^*), \tag{17}
$$

where the second inequality is owing to $k = k_{t_k}^*$ and the definition of $k_{t_k}^*$. We can rewrite Eq (17) as

$$
\begin{aligned}
\Delta_k &\le \frac{\alpha_k^{t_k}}{1-\alpha_k^{t_k}}\Big(h(S_k^*) - h(\tilde{S}_k)\Big) + 2(B_S+B_u)I_q^{t_k}(k) \\
&\le \frac{2\alpha_k^{t_k}}{1-\alpha_k^{t_k}}(B_S+B_u) + 2(B_S+B_u)I_q^{t_k}(k).
\end{aligned} \tag{18}
$$

When $\Delta_k \ge \frac{2\alpha_k^{t_k}}{1-\alpha_k^{t_k}}(B_S+B_u)$, based on the definition of the fixed confidence style confidence radius, the above inequalities can be converted to

$$
\mathcal{L}_k^\tau = \mathcal{L}_k^{t_k} \le \mathcal{N}_k^{t_k} \le \frac{8(B_S+B_u)^2\log(4K2^M t_k^2/\delta)}{\left( \Delta_k - \frac{2\alpha_k^{t_k}}{1-\alpha_k^{t_k}}(B_S+B_u) \right)^2} \le \frac{8(B_S+B_u)^2\log(4K2^M\tau^2/\delta)}{\left( \Delta_k - \frac{2\alpha_k^{t_k}}{1-\alpha_k^{t_k}}(B_S+B_u) \right)^2}, \tag{19}
$$

where the first equality follows from the definition of $t_k$, the first inequality holds because $\mathcal{N}_k^t = \sum_{s=1}^{t-1} \mathbb{1}\{k_s = k\} \ge \mathcal{L}_k^t = \sum_{s=1}^{t-1} \mathbb{1}\{k = k_s^* = k_s\}$, and the third inequality follows from $t_k \le \tau$.

Suppose

$$\mathcal{L}_k^\tau > \frac{\left((2\sqrt{2}+8)(B_S+B_u)\sqrt{\log(4K2^M\tau^2/\delta)}\right)^2}{\Delta_k^2}. \tag{20}$$

Note that under this assumption, we have $\Delta_k > \frac{2\alpha_k^\tau}{1-\alpha_k^\tau}(B_S+B_u) = \frac{2\alpha_k^{t_k}}{1-\alpha_k^{t_k}}(B_S+B_u)$ and $1 > \alpha_k^\tau = \alpha_k^{t_k}$. Substituting Eq (20) into Eq (19), we obtain

$$
\begin{aligned}
\mathcal{L}_k^\tau &\leq \frac{8(B_S+B_u)^2\log(4K2^M\tau^2/\delta)}{\left(\Delta_k - \frac{2\alpha_k^\tau}{1-\alpha_k^\tau}(B_S+B_u)\right)^2} \\
&\leq \frac{8(B_S+B_u)^2\log(4K2^M\tau^2/\delta)}{\left(\Delta_k - 4\sqrt{\frac{M}{\mathcal{L}_k^\tau}}(B_S+B_u)\right)^2} \\
&\leq \frac{\left(2\sqrt{2}(B_S+B_u)\sqrt{\log(4K2^M\tau^2/\delta)} + 4(B_S+B_u)\sqrt{M}\right)^2}{\Delta_k^2} \\
&\leq \frac{\left((2\sqrt{2}+8)(B_S+B_u)\sqrt{\log(4K2^M\tau^2/\delta)}\right)^2}{\Delta_k^2},
\end{aligned} \tag{21}
$$

where the second step follows from $\frac{\sqrt{\mathcal{L}_k^\tau}}{2} \geq \sqrt{M}$. The above inequality contradicts Eq (20), which implies that Eq (16) holds.

**Second step:** We want to prove

$$\mathcal{L}_{k^*}^\tau \leq \frac{\left((2\sqrt{2}+2)(B_S+B_u)\sqrt{\log(4K2^M\tau^2/\delta)}\right)^2}{\epsilon^2} + 1. \tag{22}$$

If $k^*$ is only triggered in the initialization phase, this result trivially holds. Besides, suppose $t_{k^*}$ is the penultimate round that arm $k$ is observed in a round that $k^* = k_{t_{k^*}}^* = k_{t_{k^*}}$. According the stopping rule (line 15 of Algorithm 1), we can derive

$$\beta_{t_{k^*}} \leq 2(B_u+B_S)I_q^{t_{k^*}}(k). \tag{23}$$

If $\beta_{t_{k^*}} > 0$, then we can substitute Eq (9) (the definition of the fixed confidence style confidence radius) into Eq (23) and derive

$$
\begin{aligned}
\mathcal{L}_{k^*}^\tau = \mathcal{L}_{k^*}^{t_{k^*}} + 1 \leq \mathcal{N}_{k^*}^{t_{k^*}} + 1 &\leq \frac{8(B_S+B_u)^2}{\beta_{t_{k^*}}^2}\log\left(\frac{4K2^M t_{k^*}^2}{\delta}\right) + 1 \\
&\leq \frac{8(B_S+B_u)^2}{\beta_{t_{k^*}}^2}\log\left(\frac{4K2^M\tau^2}{\delta}\right) + 1,
\end{aligned} \tag{24}
$$

where the first equality is owing to the definition of $t_{k^*}$ and $\mathcal{L}_{k^*}^t$ and the last inequality is owing to $t_{k^*} \leq \tau$.

Suppose

$$\mathcal{L}_{k^*}^\tau > \frac{\left((2\sqrt{2}+4)(B_S+B_u)\sqrt{\log(4K2^M\tau^2/\delta)}\right)^2}{\epsilon^2} + 1, \tag{25}$$

note that this implies $\beta_{t_{k^*}} > 0$. Recall the definition of the $\beta_t$, Eq (24) can be further expressed as

$$
\begin{aligned}
\mathcal{L}_{k^*}^{\tau} = \mathcal{L}_{k^*}^{t_{k^*}} + 1 \leq & \frac{8(B_S + B_u)^2}{\left(\epsilon - 2\alpha_{k^*}^{t_{k^*}}(B_S + B_u)\right)^2 / \left(1 - \alpha_{k^*}^{t_{k^*}}\right)^2} \log\left(\frac{4K2^M\tau^2}{\delta}\right) + 1 \\
\leq & \frac{8(B_S + B_u)^2}{\left(\epsilon - 2\alpha_{k^*}^{t_{k^*}}(B_S + B_u)\right)^2} \log\left(\frac{4K2^M\tau^2}{\delta}\right) + 1 \\
\leq & \frac{\left(2\sqrt{2}(B_S + B_u)\sqrt{\log(4K2^M\tau^2/\delta)} + 2(B_S + B_u)\sqrt{M}\right)^2}{\epsilon^2} + 1 \\
\leq & \frac{\left((2\sqrt{2} + 4)(B_S + B_u)\sqrt{\log(4K2^M\tau^2/\delta)}\right)^2}{\epsilon^2} + 1,
\end{aligned}
\tag{26}
$$

where the first inequality is owing to $0 < 1 - \alpha_{k^*}^{t_{k^*}} < 1$ and $\epsilon - 2\alpha_{k^*}^{t_{k^*}}(B_S + B_u) > 0$. Since Eq (26) contradicts Eq (25), we can finally conclude that Eq (22) holds.

**Third step:** Combine Eq (16) and Eq (22), $\tau_1$ can be bounded by

$$
\begin{aligned}
\tau_1 = & \sum_{k=1}^{K} \mathcal{L}_k^{\tau} \\
\leq & \left(96 + 48\sqrt{2}\right) H_\Delta \log\left(\frac{4K2^M\tau^2}{\delta}\right) + 1 \\
\leq & \left(96 + 48\sqrt{2}\right) H_\Delta \left(\log\left(\frac{4K\tau^2}{\delta}\right) + M\right) + 1.
\end{aligned}
$$

Note that the right-hand side of the above equation includes the $\tau$ term. In the following proof of Theorem 1, we will eliminate this term to ensure that the upper bound **does not** depend on $\tau$. This completes the proof of Lemma 3. $\qquad\square$

The proof of Lemma 3 relies on the following Lemma 7 and 8.

**Lemma 7.** *Under the event $\mathcal{E}$, we have*

$$
h(S_k^*) \leq \hat{h}_t^k \leq \left(u_k - v_{\hat{S}_{k,t}}(k)\right) + 2(B_S + B_u)I_q^t(k),
\tag{27}
$$

*for all $k \in \mathcal{A}$.*

*Proof of Lemma 7.* Recall the definition of $\hat{h}_t^k = \hat{u}_k^t + B_u I_q^t(k) - v_{LP}^*$. Based on Lemma 2, we can derive that

$$
\begin{aligned}
v_{S_k^*}(k) \geq & v_{S_k^*}(k') + C(k, k') \\
\geq & v_{S_k^*}(k') + \hat{C}^t(k, k') - I_c^t(k, k') \\
\geq & \hat{C}^t(k, k') + \hat{v}_{S_k^*}^t(k') - \left(I_c^t(k, k') + B_S I_q^t(k')\right), \quad \forall k' \neq k, t \in [\tau].
\end{aligned}
\tag{28}
$$

The first inequality follows from the definitions of $\mathcal{V}_k$ and $S_k^*$, while the second and third inequalities follow from Lemma 2. According to Eq (28), $S_k^*$ satisfies all the constraints in UCB-LP$_{k,t}$. With the definition of UCB-LP$_{k,t}$, we have

$$
\hat{u}_k^t + B_u I_q^t(k) - v_{S_k^*}(k) \leq \hat{u}_k^t + B_u I_q^t(k) - v_{LP}^* = \hat{h}_t^k.
\tag{29}
$$

Based on Lemma 2, it has

$$
u_k \leq \hat{u}_k^t + B_u I_q^t(k).
\tag{30}
$$

Combining Eq (29) and Eq (30), we finally obtain $\hat{h}_t^k > u_k - v_{S_k^*}(k) = h(S_k^*), \forall k \in \mathcal{A}$.

We will then prove the second inequality of Eq (27). There is

$$
\begin{aligned}
\hat{h}_t^k &= \hat{u}_k^t + B_u I_q^t(k) - v_{LP}^* \\
&\leq \left( \hat{u}_k^t - \hat{v}_{\hat{S}_{k,t}}^t(k) \right) + (B_S + B_u) I_q^t(k) \\
&\leq \left( u_k - v_{\hat{S}_{k,t}}(k) \right) + 2(B_S + B_u) I_q^t(k),
\end{aligned}
\tag{31}
$$

where the first inequality follows from the first constraint of UCB-LP$_{k,t}$, and the last inequality follows from Lemma 2. This completes the proof of Lemma 7. $\qquad\square$

**Lemma 8.** *Given event $\mathcal{E}$, if $k_t^* = k_t$ and $\alpha_{k_t^*}^t < 1$ in round $t$, then we have*

$$
\hat{h}_t^{k_t^*} \leq \frac{1}{1 - \alpha_{k_t^*}^t} \left( h(S_{k_t^*}^*) - \alpha_{k_t^*}^t h(\tilde{S}_{k_t^*}) \right) + 2(B_S + B_u) I_q^t(k_t^*).
\tag{32}
$$

*Proof of Lemma 8.* Remember, $k_t^* = k_t$ implies that $S_t$ can successfully induce the agent to pull $k_t^*$. According to the definition of $S_k^*$, we have

$$
\begin{aligned}
h(S_{k_t^*}^*) &\geq h(S_t) \\
&= h\left( \alpha_{k_t^*}^t \tilde{S}_{k_t^*} + (1 - \alpha_{k_t^*}^t) \hat{S}_{k_t^*, t} \right) \\
&= \alpha_{k_t^*}^t h(\tilde{S}_{k_t^*}) + (1 - \alpha_{k_t^*}^t) \left( u_{k_t^*} - v_{\hat{S}_{k_t^*}, t}(k_t^*) \right).
\end{aligned}
\tag{33}
$$

Eq (33) can be rewritten as

$$
u_{k_t^*} - v_{\hat{S}_{k_t^*}, t}(k_t^*) \leq \frac{1}{1 - \alpha_{k_t^*}^t} \left( h(S_{k_t^*}^*) - \alpha_{k_t^*}^t h(\tilde{S}_{k_t^*}) \right).
\tag{34}
$$

Based on Lemma 7, we can finally get

$$
\begin{aligned}
\hat{h}_t^{k_t^*} &\leq u_{k_t^*} - v_{\hat{S}_{k_t^*}, t} + 2(B_S + B_u) I_q^t(k_t^*) \\
&\leq \frac{1}{1 - \alpha_{k_t^*}^t} \left( h(S_{k_t^*}^*) - \alpha_{k_t^*}^t h(\tilde{S}_{k_t^*}) \right) + 2(B_S + B_u) I_q^t(k_t^*).
\end{aligned}
\tag{35}
$$

Here we finish the proof of Lemma 8. $\qquad\square$

### F.2. Proof of Lemma 4

For a binary search that concludes at round $t$, we define $t_0(t)$ as the initial round of the search, $(k_0(t), k_1(t))$ as the optimal response at the end of the search, and $(S_0(t), S_1(t))$ as the input scoring rule for this binary search.

*Proof of Lemma 4.* We can first decompose $\tau_2$ as

$$
\begin{aligned}
\tau_2 &= \sum_{t=1}^{\tau} \mathbb{1}\{k_t \neq k_t^*\} \\
&\leq \sum_{t=1}^{\tau} \left( \mathbb{1}\left\{ \mathrm{BS} = 1, \Gamma_{t_0(t)}(k_0(t), k_1(t)) \leq \alpha_{k_0(t)}^{t_0(t)} \right\} + \mathbb{1}\left\{ \mathrm{BS} = 1, \Gamma_{t_0(t)}(k_0(t), k_1(t)) > \alpha_{k_0(t)}^{t_0(t)} \right\} \right).
\end{aligned}
$$

Besides, based on Lemma 10 and Lemma 13, we know that under event $\mathcal{E}$, $\sum_{t=1}^{\tau} \mathbb{1}\{\mathrm{BS} = 1, \Gamma_{t_0(t)}(k_0(t), k_1(t)) \leq$

$\alpha_{k_0(t)}^{t_0(t)}\} = 0$ and $\sum_{t=1}^{\tau} \mathbb{1}\{\text{BS} = 1, \Gamma_{t_0(t)}(k_0(t), k_1(t)) > \alpha_{k_0(t)}^{t_0(t)}\} \leq \sum_{k=1}^{K} N_k^* \log_2(\tau)$. Therefore, we have

$$\tau_2 \leq \Big(\sum_{k=1}^{K} \alpha_k^{\tau-2}\Big)\big(12\varepsilon^{-1}B_S\big)^2 2\log_2(\tau) \log\Big(\frac{4K2^M\tau^2}{\delta}\Big)$$

$$\leq \big(96 + 48\sqrt{2}\big)\big(12\varepsilon^{-1}B_S\big)^2 2H_\Delta M^{-1} \log_2(\tau) \log^2\Big(\frac{4K2^M\tau^2}{\delta}\Big)$$

$$\leq \big(96 + 48\sqrt{2}\big)\big(12\varepsilon^{-1}B_S\big)^2 2H_\Delta M^{-1} \log_2(\tau)\Big(M + 2\log\Big(\frac{4K\tau^2}{\delta}\Big)\Big)^2.$$

Here we finish the proof of Lemma 4. □

The proof of Lemma 4 mainly relies on Lemma 10 and 13. To prove Lemma 10, we need to invoke the following Lemma 9.

**Lemma 9.** *We assume that $\Gamma_t(k_0, k_1) \leq \alpha_{k_0}^t$ and $k_0 \neq k_1$. Under the event $\mathcal{E}$, the agent will not respond by pulling arm $k_1$ according to the scoring rule $S_t = \alpha_{k_0}^t \tilde{S}_{k_0} + (1 - \alpha_{k_0}^t)\hat{S}_{k_0,t}$.*

*Proof of Lemma 9.* Under the scoring rule $S_t = \alpha_{k_0}^t \tilde{S}_{k_0} + (1 - \alpha_{k_0}^t)\hat{S}_{k_0,t}$ and the event $\mathcal{E}$, the expected profit for the agent generated by responding with arm $k_1$ satisfies

$$v_{\alpha_{k_0}^t \tilde{S}_{k_0} + (1-\alpha_{k_0}^t)\hat{S}_{k_0,t}}(k_1) - c_{k_1} \leq \alpha_{k_0}^t v_{\tilde{S}_{k_0}}(k_1) + (1 - \alpha_{k_0}^t)\hat{v}_{\hat{S}_{k_0,t}}^t(k_1) - c_{k_1} + (1 - \alpha_{k_0}^t)B_S I_q^t(k_1),$$

and the expected profit generated by arm $k_0$ satisfies

$$v_{\alpha_{k_0}^t \tilde{S}_{k_0} + (1-\alpha_{k_0}^t)\hat{S}_{k_0,t}}(k_0) - c_{k_0} \geq \alpha_{k_0}^t v_{\tilde{S}_{k_0}}(k_0) + (1 - \alpha_{k_0}^t)\hat{v}_{\hat{S}_{k_0,t}}^t(k_0) - c_{k_0} - (1 - \alpha_{k_0}^t)B_S I_q^t(k_0).$$

By the action-informed oracle assumption, we already have

$$v_{\tilde{S}_{k_0}}(k_0) - c_{k_0} - v_{\tilde{S}_{k_0}}(k_1) + c_{k_1} \geq \varepsilon.$$

Since $\hat{S}_{k_0,t}$ is the solution to UCB-LP$_{k_0,t}$, following the last constraint of UCB-LP$_{k_0,t}$, we obtain

$$\hat{v}_{\hat{S}_{k_0,t}}^t(k_0) - \hat{v}_{\hat{S}_{k_0,t}}^t(k_1) \geq \hat{C}^t(k_0, k_1) - I_c^t(k_0, k_1) - B_S(I_q^t(k_0) + I_q^t(k_1)). \tag{36}$$

Combining the above inequalities, we obtain

$$v_{\alpha_{k_0}^t \tilde{S}_{k_0} + (1-\alpha_{k_0}^t)\hat{S}_{k_0,t}}(k_0) - c_{k_0} - v_{\alpha_{k_0}^t \tilde{S}_{k_0} + (1-\alpha_{k_0}^t)\hat{S}_{k_0,t}}(k_0) - c_{k_0}$$
$$\geq \alpha_{k_0}^t\Big(v_{\tilde{S}_{k_0}}(k_0) - c_{k_0} - v_{\tilde{S}_{k_0}}(k_1) + c_{k_1}\Big) + (1 - \alpha_{k_0}^t)\Big(\hat{v}_{\hat{S}_{k_0,t}}^t(k_0) - \hat{v}_{\hat{S}_{k_0,t}}^t(k_1) - C(k_0, k_1)\Big)$$
$$\quad - (1 - \alpha_{k_0}^t)B_S\big(I_q^t(k_0) + I_q^t(k_1)\big)$$
$$\geq \alpha_{k_0}^t \varepsilon + (1 - \alpha_{k_0}^t)\Big(\hat{C}^t(k_0, k_1) - C(k_0, k_1) - I_c^t(k_0, k_1) - B_S\big(I_q^t(k_0) + I_q^t(k_1)\big)\Big)$$
$$\quad - (1 - \alpha_{k_0}^t)B_S\big(I_q^t(k_0) + I_q^t(k_1)\big)$$
$$\geq \alpha_{k_0}^t \varepsilon + 2(1 - \alpha_{k_0}^t)\Big(-I_c^t(k_0, k_1) - B_S\big(I_q^t(k_0) + I_q^t(k_1)\big)\Big)$$
$$\geq \alpha_{k_0}^t \varepsilon + 2\Big(-I_c^t(k_0, k_1) - B_S\big(I_q^t(k_0) + I_q^t(k_1)\big)\Big).$$

When $\alpha_{k_0}^t \geq 2\varepsilon^{-1}\Big(I_c^t(k_0, k_1) + B_S\big(I_q^t(k_0) + I_q^t(k_1)\big)\Big)$, it follows that $v_{\tilde{S}_{k_0}}(k_0) - c_{k_0} - v_{\tilde{S}_{k_0}}(k_1) + c_{k_1} \geq 0$, which means that the agent prefers to respond by taking arm $k_0$ rather than arm $k_1$. This completes the proof of Lemma 9. □

Based on Lemma 9, we can directly prove Lemma 10.

**Lemma 10.** *Under the event $\mathcal{E}$, we have $\sum_{t=1}^{\tau} \mathbb{1}\{BS = 1, \Gamma_{t_0(t)}(k_0(t), k_1(t)) \leq \alpha_{k_0(t)}^{t_0(t)}\} = 0$.*

*Proof of Lemma 10.* When conducting a binary search, the scoring rule $S_t$ falls within the segment $(S_0(t), S_1(t))$ (refer to Algorithm 3). It's worth noting that $S_0(t) = \alpha_{k_0(t)}^{t_0(t)} \tilde{S}_{k_{t_0(t)}^*} + (1 - \alpha_{k_0(t)}^{t_0(t)}) \hat{S}_{k_{t_0(t)}^*, t_0(t)}$ and $S_1(t) = \tilde{S}_{k_{t_0(t)}^*}$. Therefore, the scoring rule during the binary search can be represented as

$$S_t = \alpha' \tilde{S}_{k_{t_0(t)}^*} + (1 - \alpha') \hat{S}_{k_{t_0(t)}^*, t_0(t)}, \tag{37}$$

where $\alpha_{k_0(t)}^{t_0(t)} \leq \alpha' \leq 1$. Hence, under assumption $\Gamma_{t_0(t)}(k_0(t), k_1(t)) \leq \alpha_{k_0(t)}^{t_0(t)}$, the inequality $\Gamma_{t_0(t)}(k_0(t), k_1(t)) \leq \alpha'$ also holds. We now consider two different scenarios.

The first scenario arises during the binary search (from $t_0(t) + 1$ to $t_1(t)$), where the agent only responds by taking arm $k_1(t) = k_{t_0(t)}^*$. This implies that $k_0(t) = k_{t_0(t)}$. However, the condition $k_0(t) \neq k_1(t)$, together with the assumption $\Gamma_{t_0(t)}(k_0(t), k_1(t)) \leq \alpha_{k_0(t)}^{t_0(t)}$ and Lemma 9, implies that the agent cannot respond by taking $k_0(t)$ in the $t_0(t)$ round, which contradicts the definition of this scenario.

The second scenario emerges during the binary search, where the agent responds by taking an arm not equal to $k_1(t)$ at least once. However, given that $k_0(t) \neq k_1(t)$, the assumption $\Gamma_{t_0(t)}(k_0(t), k_1(t)) \leq \alpha_{k_0(t)}^{t_0(t)} \leq \alpha'$, and Lemma 9, the agent will not respond by taking $k_0(t)$ during the binary search. This also contradicts the definition of the scenario.

The above discussion implies that in every binary search, $\Gamma_{t_0(t)}(k_0(t), k_1(t)) > \alpha_{k_0(t)}^{t_0(t)}$. Therefore, under the event $\mathcal{E}$, we have $\sum_{t=1}^{\tau} \mathbb{1}\{\mathrm{BS} = 1, \Gamma_{t_0(t)}(k_0(t), k_1(t)) \leq \alpha_{k_0(t)}^{t_0(t)}\} = 0$. Here we finish the proof of Lemma 13. $\qquad\square$

We now utilize Lemma 11, 12, and 13 to show that $\sum_{t=1}^{\tau} \mathbb{1}\{\mathrm{BS} = 1, \Gamma_{t_0(t)}(k_0(t), k_1(t)) > \alpha_{k_0(t)}^{t_0(t)}\} \leq \sum_{k=1}^{K} N_k^* \log_2(\tau)$. Lemma 11 and Lemma 12 demonstrate how binary search can decrease $I_c^t$ and establish the stopping criteria for essential binary search, respectively.

**Lemma 11.** *For a binary search $BS(S_0, S_1, \tilde{\lambda}_{\min}, \tilde{\lambda}_{\max}, k_0, k_1, t_0, t_1)$, we have*

$$I_c^t(k_0, k_1) \leq 2\sqrt{2\log\left(\frac{4K2^M \tau^2}{\delta}\right)} B_S \left(\frac{1}{\sqrt{\mathcal{N}_{k_0}^{t_1}}} + \frac{1}{\sqrt{\mathcal{N}_{k_1}^{t_1}}}\right).$$

*Proof of Lemma 11.* The proof of Lemma 11 directly follows the proof of Proposition F.9 in Chen et al. (2023). $\qquad\square$

**Lemma 12.** *For a binary search $BS(S_0, S_1, \tilde{\lambda}_{\min}, \tilde{\lambda}_{\max}, k_0, k_1, t_0, t_1)$, if event $\mathcal{E}$ happens and*

$$\min\{\mathcal{N}_{k_0}^{t_1}, \mathcal{N}_{k_1}^{t_1}\} \geq \left(12B_S(\alpha_{k_0}^{\tau}\varepsilon)^{-1}\right)^2 2\log\left(\frac{4K2^M \tau^2}{\delta}\right) = N_{k_0}^*, \tag{38}$$

*then we have $\alpha_{k_0}^t \geq \Gamma_t(k_0, k_1)$ for $t_1 < t \leq \tau$, and no more essential binary search ending with the same $(k_0, k_1)$ is possible.*

*Proof of Lemma 12.* For any $t_1 < t \leq \tau$, we have

$$
\begin{aligned}
\alpha_{k_0}^t &\geq \alpha_{k_0}^{\tau} \\
&= \left(12\varepsilon^{-1}B_S\right)\sqrt{\frac{2\log\left(4K2^M\tau^2/\delta\right)}{N_{k_0}^*}} \\
&\geq \left(2\varepsilon^{-1}B_S\right)\sqrt{2\log\left(\frac{4K2^M\tau^2}{\delta}\right)}\frac{6}{\min\{\mathcal{N}_{k_0}^{t_1}, \mathcal{N}_{k_1}^{t_1}\}} \\
&\geq \left(2\varepsilon^{-1}B_S\right)\sqrt{2\log\left(\frac{4K2^M\tau^2}{\delta}\right)}\left(2\frac{1}{\sqrt{\mathcal{N}_{k_0}^{t_1}}} + 2\frac{1}{\sqrt{\mathcal{N}_{k_1}^{t_1}}} + \frac{1}{\sqrt{\mathcal{N}_{k_0}^t}} + \frac{1}{\sqrt{\mathcal{N}_{k_1}^t}}\right) \\
&\geq 2\varepsilon^{-1}\left(I_c^t(k_0, k_1) + B_S\left(I_q^t(k_0) + I_q^t(k_1)\right)\right) \\
&= \Gamma_t(k_0, k_1),
\end{aligned}
$$

where the first inequality is owing to $\alpha_k^t$ is is a parameter that is monotonically non-increasing over time, the first equality is due to the definition of $N_{k_0}^*$, the first inequality is due to Eq (38), the second inequality is due to $\mathcal{N}_k^t \geq \mathcal{N}_k^{t_1}, \forall k \in \mathcal{A}$, and the third inequality is owing to Lemma 11. Therefore, we conclude that after round $t_1$, $\alpha_{k_0}^t$ is larger than $\Gamma_t(k_0, k_1)$ and no more essential binary search will ending with $(k_0, k_1)$. Here we finish the proof of Lemma 12. $\qquad\square$

**Lemma 13.** *A binary search* $BS(S_0, S_1, \tilde{\lambda}_{\min}, \tilde{\lambda}_{\max}, k_0, k_1, t_0, t_1)$ *is an essential binary search if and only if* $\Gamma_{t_0}(k_0, k_1) > \alpha_{k_0}^{t_0}$. *The total number of essential binary searches in* $\tau$ *rounds can be bounded by* $\sum_{k=1}^K N_k^* m \leq \sum_{k=1}^K N_k^* \log_2(\tau) = \sum_{k=1}^K \left(12 B_S (\alpha_k^\tau \varepsilon)^{-1}\right)^2 2 \log_2(\tau) \log\left(\frac{4K2^M \tau^2}{\delta}\right)$.

*Proof of Lemma 13.* The result directly follows the proof of Lemma F.4 in Chen et al. (2023) and Lemma 12. $\qquad\square$

### F.3. Proof of Lemma 5

*Proof of Lemma 5.* We have

$$
\begin{aligned}
2(B_u + B_S) I_q^\tau(k_\tau^*) &\geq \hat{h}_\tau^{k^*} - \hat{h}_\tau^{k_\tau^*} + 2(B_u + B_S) I_q^\tau(k_\tau^*) \\
&\geq \hat{h}_\tau^{k^*} - u_{k_\tau^*} + v_{\hat{S}_{k_\tau^*, \tau}}(k_\tau^*) - 2(B_u + B_S) I_q^\tau(k_\tau^*) + 2(B_u + B_S) I_q^\tau(k_\tau^*) \\
&\geq h(S^*) - u_{k_\tau^*} + v_{\hat{S}_{k_\tau^*, \tau}}(k_\tau^*),
\end{aligned}
\tag{39}
$$

where the first inequality follows from the definition of $k_t^*$, and the second and third inequalities follow from Lemma 7. According to the definition of the termination round $\tau$, $\alpha_{k_\tau^*}^\tau$ must be less than 1; otherwise, $\beta_\tau = 0$ cannot exceed $2(B_S + B_u) I_q^\tau(k_\tau^*)$, since the latter term is strictly positive. We have

$$
h(S^*) - u_{k_\tau^*} + v_{\hat{S}_{k_\tau^*, \tau}}(k_\tau^*) \leq \beta_\tau = \frac{\epsilon - 2\alpha_{k_\tau^*}^\tau (B_S + B_u)}{1 - \alpha_{k_\tau^*}^\tau}.
$$

Moreover, since $\hat{S}^* = S_\tau = \alpha_{k_\tau^*}^\tau \tilde{S}_{k_\tau^*} + (1 - \alpha_{k_\tau^*}^\tau) \hat{S}_{k_\tau^*, \tau}$ and the best response to $\hat{S}^*$ is $k_\tau^*$, we have

$$
\begin{aligned}
h(S^*) - h(\hat{S}^*) &= h(S^*) - u_{k_\tau^*} + v_{\alpha_{k_\tau^*}^\tau \tilde{S}_{k_\tau^*} + (1 - \alpha_{k_\tau^*}^\tau) \hat{S}_{k_\tau^*, \tau}}(k_\tau^*) \\
&= \alpha_{k_\tau^*}^\tau \left( h(S^*) - h(\tilde{S}_{k_\tau^*}) \right) + (1 - \alpha_{k_\tau^*}^\tau) \left( h(S^*) - u_{k_\tau^*} + v_{\hat{S}_{k_\tau^*, \tau}}(k_\tau^*) \right) \\
&\leq 2\alpha_{k_\tau^*}^\tau (B_S + B_u) + (1 - \alpha_{k_\tau^*}^\tau) \left( h(S^*) - u_{k_\tau^*} + v_{\hat{S}_{k_\tau^*, \tau}}(k_\tau^*) \right) \\
&\leq 2\alpha_{k_\tau^*}^\tau (B_S + B_u) + (1 - \alpha_{k_\tau^*}^\tau) \beta_\tau \\
&= \epsilon,
\end{aligned}
\tag{40}
$$

where the last equality is owing to the definition of $\beta_\tau$.

Note that $\mathcal{E}$ happens with probability at least $1 - \delta$ (as shown in Lemma 2), we can conclude that $\hat{S}^*$ can satisfy the $(\epsilon, \delta)$-condition (3). Here we finish the proof of Lemma 5. $\qquad\square$

### F.4. Proof of Theorem 1

*Proof of Theorem 1.* Lemma 5 directly shows that the estimated optimal scoring rule $\hat{S}^*$ satisfies condition (3). Here, we present the upper bound of the sample complexity. Recalling that $\tau = \tau_1 + \tau_2$, and combining Lemma 3 and Lemma 4, we obtain

$$
\tau \leq \left( (96 + 48\sqrt{2}) H_\Delta \left( M + \log\left(\frac{4K\tau^2}{\delta}\right) \right) + 1 \right) \left( 1 + \left( 12\varepsilon^{-1} B_S \right)^2 2 \log_2(\tau) M^{-1} \left( M + \log\left(\frac{4K\tau^2}{\delta}\right) \right) \right).
\tag{41}
$$

We define $c = \left(97 + 48\sqrt{2}\right)2$, Eq (41) can be rewritten as

$$
\begin{aligned}
\tau \leq & c\left(1 + 12\varepsilon^{-1}B_S\right)^2 H_\Delta \log_2(\tau) M^{-1}\left(M + \log\left(\frac{4K\tau^2}{\delta}\right)\right)^2 \\
= & c\left(1 + 12\varepsilon^{-1}B_S\right)^2 H_\Delta \log_2(\tau)\left(M + 2\log\left(\frac{4K\tau^2}{\delta}\right) + \frac{1}{M}\log^2\left(\frac{4K\tau^2}{\delta}\right)\right).
\end{aligned}
\tag{42}
$$

Define a parameter $\Lambda$, such that

$$
\begin{aligned}
\Lambda \leq \tau = & c\left(1 + 12\varepsilon^{-1}B_S\right)^2 H_\Delta \log_2(\Lambda) M^{-1}\left(M + \log\left(\frac{4K\Lambda^2}{\delta}\right)\right)^2 \\
\leq & c\left(1 + 12\varepsilon^{-1}B_S\right)^2 H_\Delta \log_2(\Lambda) M^{-1}\left(M + \log\left(\left(\frac{4K\Lambda}{\delta}\right)^2\right)\right)^2 \\
\leq & 128c\left(1 + 12\varepsilon^{-1}B_S\right)^2 H_\Delta \log_2\left((\Lambda)^{1/2}\right)\left(M + \log\left(\left(\frac{4K\Lambda}{\delta}\right)^{1/4}\right)\right)^2 \\
\leq & 128c\left(1 + 12\varepsilon^{-1}B_S\right)^2 H_\Delta\left(\Lambda^{1/4}M^2 + 2M\left(\frac{4K\Lambda}{\delta}\right)^{3/8} + \left(\frac{4K\Lambda}{\delta}\right)^{1/2}\right) \\
\leq & \left(128c\left(1 + 12\varepsilon^{-1}B_S\right)^2 H_\Delta\left(M^2 + 2M\left(\frac{4K}{\delta}\right)^{3/8} + \left(\frac{4K}{\delta}\right)^{1/2}\right)\right)^2 \\
= & \Gamma,
\end{aligned}
\tag{43}
$$

where the third inequality is owing to $\sqrt{x} \geq \log_2(x) \geq \log(x)$ for all $x > 0$. Substituting Eq (43) into Eq (41), we can finally derive

$$
\begin{aligned}
\tau \leq & c\left(1 + 12\varepsilon^{-1}B_S\right)^2 H_\Delta \log_2(\Gamma)\left(M + 2\log\left(\frac{4K\Gamma^2}{\delta}\right) + \frac{1}{M}\log^2\left(\frac{4K\Gamma^2}{\delta}\right)\right) \\
= & \tilde{O}\left(B_S^2\varepsilon^{-2}MH_\Delta\right).
\end{aligned}
\tag{44}
$$

Note that the right-hand side of the above upper bound does not contain $\tau$. Here we finish the proof of Theorem 1. $\quad\square$

# G. Proof of Corollary 1

Following a similar approach as in the previous section, we split this analysis into three parts. First, we will establish an upper bound for $\tau_1 = \sum_{t=1}^{\tau} \mathbb{1}\{k_t = k_t^*\}$. Then, in the second part, we will derive an upper bound for $\tau_2 = \sum_{t=1}^{\tau} \mathbb{1}\{k_t \neq k_t^*\}$. Finally, by summing the upper bounds of $\tau_1$ and $\tau_2$, and applying some basic algebraic manipulations, we obtain the upper bound for $\tau$. We will also demonstrate that, with the parameters specified in Corollary 1, the estimated best arm satisfies condition (3).

**Upper bound $\tau_1$**    Under event $\mathcal{E}$, we want to prove

$$\mathcal{L}_k^{\tau} \leq \frac{16(B_S + B_u)^2}{\epsilon^2} \log\left(\frac{4K2^M\tau^2}{\delta}\right) + K, \ \forall k \in \mathcal{A}. \tag{45}$$

Suppose arm $k \in \mathcal{A}$ is only triggered during the initialization phase; in this case, the result holds trivially. Furthermore, let $t_k$ represent the penultimate round in which arm $k$ is observed, such that $k^* = k_{t_k}^* = k_{t_k}$. Based on the stopping rule (line 15 of Algorithm 1), we can derive that

$$\beta_{t_k} \leq 2(B_u + B_S)I_q^{t_k}(k). \tag{46}$$

Note that based on the setting in Corollary 1, it has $0 < \beta_t$ for all $t$. We can substitute Eq (9) (the definition of the fixed confidence style confidence radius) into Eq (46) and derive

$$
\begin{aligned}
\mathcal{L}_k^{\tau} = \mathcal{L}_k^{t_k} + 1 \leq \mathcal{N}_k^{t_k} + 1 &\leq \frac{4(B_S + B_u)^2}{\beta_{t_k}^2} \log\left(\frac{4K2^M t_k^2}{\delta}\right) + 1 \\
&\leq \frac{4(B_S + B_u)^2}{\beta_{t_k}^2} \log\left(\frac{4K2^M\tau^2}{\delta}\right) + 1 \\
&\leq \frac{16(B_S + B_u)^2}{\epsilon^2} \log\left(\frac{4K2^M\tau^2}{\delta}\right) + 1,
\end{aligned} \tag{47}
$$

where the first equality is owing to the definition of $t_k$ and $\mathcal{L}_k^t$, the third inequality is owing to $t_k \leq \tau$ and the last inequality is owing to $\beta_{t_k} \geq \frac{\epsilon}{2}$. With the last term in Eq (47), we can derive Eq (45), and

$$
\begin{aligned}
\tau_1 = \sum_{k=1}^{K} \mathcal{L}_k^{\tau} &\leq \frac{16(B_S + B_u)^2 K}{\epsilon^2} \log\left(\frac{4K2^M\tau^2}{\delta}\right) + K \\
&= 4H_\epsilon \log\left(\frac{4K2^M\tau^2}{\delta}\right) + K \\
&\leq 4H_\epsilon\left(M + \log\left(\frac{4K\tau^2}{\delta}\right)\right) + K.
\end{aligned} \tag{48}
$$

**Upper bound $\tau_2$**    Based on the discussion in the previous section, if $\mathcal{E}$ happens, we have

$$
\begin{aligned}
\tau_2 &\leq \sum_{k=1}^{K} N_k^* m \\
&= \sum_{k=1}^{K} \left(12B_S\left(\alpha_k^\tau \varepsilon\right)^{-1}\right)^2 2\log_2(\tau) \log\left(\frac{4K2^M\tau^2}{\delta}\right) \\
&= 1152H_\epsilon B_S^2 \varepsilon^{-2} \log_2(\tau)\left(M + \log\left(\frac{4K\tau^2}{\delta}\right)\right).
\end{aligned} \tag{49}
$$

**Upper bound $\tau$ and demonstrate $\hat{S}^*$ satisfies Eq (3)**    Note that the event $\mathcal{E}$ occurs with probability at least $1 - \delta$. By combining Eq (45) and Eq (49) and following the steps outlined in the previous section, we conclude that the following holds with probability at least $1 - \delta$:

$$\tau = \tilde{O}\left(B_S^2 \varepsilon^{-2} M H_\epsilon\right). \tag{50}$$

Furthermore, following the same reasoning as in the proof of Lemma 5, we can show that $\hat{S}^*$ satisfies Eq (3). This concludes the proof of Corollary 1

# H. Proof of Theorem 2

**Proof sketch of Theorem 2** In Lemma 14, we demonstrate that OIAFB will output an estimated best scoring rule if and only if $0 < a < \frac{T}{144H_\epsilon B_S^2 \varepsilon^{-2} \log_2(T)}$. Furthermore, we show that, within the event $\mathcal{E}$, if the simple regret $h(S^*) - h(\hat{S}^*)$ exceeds $\epsilon$, then $T \leq 4aH_\epsilon + K$. It follows that selecting $0 < a < \min\left(\frac{T-K}{4H_\epsilon}, \frac{T}{144H_\epsilon B_S^2 \varepsilon^{-2} \log_2(T)}\right)$ leads to $T > 4H_\epsilon a + K$, which contradicts our previous findings. Consequently, within the event $\mathcal{E}$, OIAFB will yield an estimated scoring rule with a simple regret no larger than $\epsilon$. Additionally, Lemma 2 indicates that the event $\mathcal{E}$ occurs with a probability of at least $1 - TK2^M e^{-\frac{1}{2}a}$ when adopting the confidence radius defined in Eq (10).

**Lemma 14.** *Suppose $0 < a < \frac{T}{144H_\epsilon B_S^2 \varepsilon^{-2} \log_2(T)}$. Under event $\mathcal{E}$, if $h(S^*) - h(\hat{S}^*) > \epsilon$, then we have $T \leq 4H_\epsilon a + K$.*

*Proof of Lemma 14.* Note that OIAFB will return an estimated best scoring rule if and only if $\sum_{t=1}^T \mathbb{1}\{k_t = k_t^*\}$ is greater than 1 (see Algorithm 2 for details). When we select $I_q^t$ as defined in Eq (10), $\sum_{t=1}^T \mathbb{1}\{k_t \neq k_t^*\}$ can be bounded by $144H_\epsilon B_S^2 \varepsilon^{-2} a \log_2(T)$ (similarly to Lemma 4 and using $m \leq \log_2(T)$ in the fixed-budget literature). Thus, to ensure that OIAFB outputs an estimated best arm, $a$ must satisfies $0 < a < \frac{T}{144H_\epsilon B_S^2 \varepsilon^{-2} \log_2(T)}$.

We want to show $\mathcal{L}_k^T \leq 4H_\epsilon a + 1$, for all $k \in \mathcal{A}$. If arm $k$ is only triggered in the initial phase, this inequality trivially holds. Otherwise, define $t_k$ as the penultimate round that arm $k$ is triggered by the agent and $k_t = k_t^*$. It has

$$2(B_S + B_u)I_q^{t_k}(k) \geq 2(B_S + B_u)I_q^{t^*}(\hat{k}^*) \geq h(S^*) - u_{\hat{k}^*} + v_{\hat{S}^*}(\hat{k}^*) \geq \frac{\epsilon - 2\alpha(B_S + B_u)}{1 - \alpha_k^t} = \beta_t, \tag{51}$$

where $\hat{k}^*$ is the best response of $\hat{S}^*$. The first inequality of Eq (51) is owing to the definition of $\hat{k}^*$ and $\hat{t}^*$, the second inequality is based on Eq (39), and the third inequality is owing to

$$\begin{aligned}
h(S^*) - u_{\hat{k}^*} + v_{\hat{S}^*}(\hat{k}^*) &= \frac{h(S^*) - h(\hat{S}^*) - \alpha_k^t\left(h(S^*) - h(\tilde{S}_{k_\tau})\right)}{1 - \alpha_k^t} \\
&\geq \frac{h(S^*) - h(\hat{S}^*) - 2\alpha_k^t(B_S + B_u)}{1 - \alpha_k^t} \\
&\geq \frac{\epsilon - 2\alpha_k^t(B_S + B_u)}{1 - \alpha_k^t},
\end{aligned} \tag{52}$$

where the first equality is based on the second equality of Eq (40). Based on Eq (51), the definition of fixed budget style $I_q^t(k)$ and the definition of $t_k$, it is easy to derive

$$\mathcal{L}_k^T = \mathcal{L}_k^{t_k} + 1 \leq \mathcal{N}_k^{t_k} + 1 \leq \frac{4(B_S + B_u)^2 a}{\beta_t^2} + 1, \quad \forall k \in \mathcal{A}. \tag{53}$$

This implies

$$T_1 = \sum_{k=1}^K \mathcal{L}_k^T \leq \frac{4K(B_S + B_u)^2 a}{\beta_t^2} + K. \tag{54}$$

According to the setting that $\alpha_k^t = \frac{\epsilon}{4(B_S + B_u)}$, we have $\beta_t \geq \frac{\epsilon}{2}$. This means

$$T \leq 4H_\epsilon a + K. \tag{55}$$

Here we finish the proof of Lemma 14. $\square$

*Proof of Theorem 2.* If we set $2\log\left(TK2^M\right) \leq a < \min\left(\frac{T-K}{4H_\epsilon}, \frac{T}{144H_\epsilon B_S^2 \varepsilon^{-2} \log_2(T)}\right)$, it indicates that $4H_\epsilon a + K < T$, which contradicts the previous result. Therefore, under event $\mathcal{E}$, if $2\log\left(TK2^M\right) \leq a < \frac{T-K}{144H_\epsilon \max\left(B_S^2 \varepsilon^{-2}, 1\right) \log_2(T)} \leq$ $\min\left(\frac{T-K}{4H_\epsilon}, \frac{T}{144H_\epsilon B_S^2 \varepsilon^{-2} \log_2(T)}\right)$, then $h(S^*) - h(\hat{S}^*) \leq \epsilon$. According to Lemma 2, $\mathcal{E}$ will happen with probability at least $1 - TK2^M e^{-\frac{1}{2}a}$ if we set $I_q^t$ as Eq (10). This implies $\tilde{\delta} \leq TK2^M e^{-\frac{1}{2}a} \leq 1$. This completes the proof of Theorem 2. $\square$

# I. Proof of Corollay 2

Due to $a = 2\log\left(TK2^M/\delta\right)$ should be smaller than $\frac{T-K}{144H_\epsilon \max\left(B_S^2\varepsilon^{-2},1\right)\log_2(T)}$, $T$ should satisfies

$$T > 288H_\epsilon \max\left(B_S^2\varepsilon^{-2},1\right)\log_2(T)\left(M + \log\left(\frac{TK}{\delta}\right)\right) + K. \tag{56}$$

We define $\Gamma = \left(1152H_\epsilon \max\left(B_S^2\varepsilon^{-2},1\right)\left(M + (K/\delta)^{1/2}\right) + K\right)^2$, then

$$
\begin{aligned}
\Gamma =& \left(1152H_\epsilon \max\left(B_S^2\varepsilon^{-2},1\right)\left(M + (K/\delta)^{1/2}\right) + K\right)^2 \\
\geq& 1152H_\epsilon \max\left(B_S^2\varepsilon^{-2},1\right)\left(\Gamma^{1/4}M + (\Gamma K/\delta)^{1/2}\right) + K \\
\geq& 1152H_\epsilon \max\left(B_S^2\varepsilon^{-2},1\right)\log_2\left(\Gamma^{1/2}\right)\left(M + \log\left((\Gamma K/\delta)^{1/2}\right)\right) + K \\
>& 288\max\left(B_S^2\varepsilon^{-2},1\right)\log_2(\Gamma)\left(M + \log\left(\Gamma K/\delta\right)\right) + K,
\end{aligned}
\tag{57}
$$

where the first inequality is owing to $\log(x) \leq \log_2(x) \leq \sqrt{x}$ for all $x > 1$. If

$$T = 288H_\epsilon \max\left(B_S^2\varepsilon^{-2},1\right)\log_2(\Gamma)\left(M + \log\left(\Gamma K/\delta\right)\right) + K, \tag{58}$$

then, we have $\Gamma > T$, and Eq (56) trivially holds. Finally, by substituting $a = 2\log\left(TK2^M/\delta\right)$ into $\tilde{\delta} \leq TK2^M e^{-\frac{1}{2}a}$, we obtain $\tilde{\delta} \leq \delta$.

