# OpenReview forum: "Provably Efficient Algorithm for Best Scoring Rule Identification in Online Principal-Agent Information Acquisition"
_ICML.cc/2025/Conference — ICML 2025 poster_

### Official Review · Reviewer_ksDs · 2025-03-12

**Overall Recommendation:** 3

**Summary:**

This paper studies a principal-agent problem, where the principal aims at learning the optimal scoring rule for that problem. Notably, it proposes pure exploration algorithms in both settings of fixed budget and fixed confidence

**Claims And Evidence:**

See below

**Essential References Not Discussed:**

See below

**Experimental Designs Or Analyses:**

See below

**Methods And Evaluation Criteria:**

See below

**Other Comments Or Suggestions:**

The problem of learning optimal pricing/scoring rules in principal agent is interesting and has been widely considered in the literature, as suggested by the related work section of this paper.

My main concern about this work is about the (over)complexity of the setting. I indeed find it quite overcomplicated, while similar ideas and messages can be found with much simpler problems. Notably, I fear that the Assumption 1, along with the fact that the agent plays before observing the nature state (which leads to Equation 5), makes this problem nearly equivalent to much simpler settings considered in the literature, such as the mentioned work of Scheid et al (2024). While the setting should be more intricate here, the Assumption 1 drastically simplifies the setting, ending up in something close to Scheid et al (2024), but in a pure exploration setting. If that is indeed the case, I would have just preferred an adaptation of this simpler setting to pure exploration.

This overcomplexity also makes the paper unnecessarily unclear: there are a lot of technical notations and considerations and it is hard to read through. I really think that the use of a simpler, but morally similar setting, would clearly help in that direction.

Also, the agent is here assumed to be myopic, i.e., they only react optimally to the current scoring rule. An agent could be nastier in reality, manipulating the learning of the principal so that it can maximize their own reward. Moreover, the agents are even assumed to be truthful, based on the existence of proper scoring rules.
While Myerson guarantees the existence of an optimal proper rule, how easy is it to transform any scoring rule into a proper one? This would probably require some knowledge of the game environment, that the principal does not have at the beginning of the game here (+ a potentially intensive computation).

----------

Here are more minor remarks/comments:
- it is not clear what is the exact knowledge of the principal? Notably for the UCB-LP formulation, it seems that knowledge of the outcome probability $p$ is necessary
- what is the value of B_S in Lemma 1 ? I guess it cannot be made arbitrarily small
- How computationally intense is the solving of the LP programming in UCB-LP?
- line 310: if $k_t=k_t^*$, this could also mean the principal is overpaying, which is not something we are happy about

**Other Strengths And Weaknesses:**

See below

**Questions For Authors:**

see above

**Relation To Broader Scientific Literature:**

See below

**Theoretical Claims:**

See below

---

> ### Author Rebuttal · Authors · 2025-04-01
>
> Thank you for your valuable feedback, which has greatly contributed to improving our work.
>
> **Q1:** My main concern about this work is about the (over)complexity of the setting ... help in that direction.
>
> **A1:** Thank you for your insightful comment. We understand the concern about complexity, and we’d like to clarify why our framework is intentionally more involved than works like Scheid (2024).
>
> The key difference from Scheid (2024) lies in the presence of an underlying state in our setting. The principal in our setting seeks to acquire high-quality information about this hidden state through the agent’s report, as the report quality and the underlying state-generating process have a direct impact on the principal’s profit. To achieve this, the principal designs a scoring rule that incentivizes the agent to conduct a good investigation (action) and report truthfully. This setup closely mirrors real-world scenarios—such as a manager hiring a domain expert to assess a situation—where **the profit of the principal depends on the true state and the quality of the report, not just the agent’s action**.
>
> In contrast, the setting of Scheid (2024) does not involve an underlying state—the principal’s objective is to induce a desirable action via a payment rule. While this is suitable for scenarios where **the principal’s profit depends directly on the agent’s action**, it is less appropriate for richer interactions such as the manager–expert example introduced earlier.
>
> Another advantage of our setting is its generality—it subsumes several important principal–agent problems as special cases, including the contract design problem (see the discussion in Chen (2023), Appendix B).
>
> **Q2:** Also, the agent is here assumed to be myopic... Moreover, the agents are even assumed to be truthful...
>
> **A2:** Thank you for your comment. Assuming the agent is myopic is a standard modeling choice in famous online principal–agent problems (e.g., Zhu 2022; Chen 2023; Scheid 2024). This is mainly because, in practice, the agent often lacks sufficient information about the principal’s future policy, making it difficult for them to plan long-term strategies that maximize their own profit. We agree that relaxing the myopic assumption and studying strategic, far-sighted agents is an interesting direction for future work.
>
> Additionally, we would like to clarify that our framework focuses solely on proper scoring rules. By the revelation principle, this restriction will not result in any loss of optimality in terms of the principal’s expected profit [Chen (2023), Cacciamani et al. (2023)]. Therefore, our goal is to identify an optimal rule within the set of proper scoring rules that maximizes the principal’s expected profit. Thus we do not need to transform arbitrary scoring rules into proper ones.
>
> **Q3:** It is not clear what is the ... $p$ is necessary.
>
> **A3:** Thank you very much for your question. The principal is assumed to know its own utility function $u$. We respectfully notice that in the UCB-LP, there is no parameter named $p$. If your question refers to $q$, which denotes the distribution over the agent’s belief, then indeed the principal does not know it. Instead, the principal must gradually learn this belief distribution over time based on interactions with the agent (see Section 4.2).
>
> **Q4:** What is the value of $B_S$ in Lemma 1 ?
>
> **A4:** The constant $B_S$ is a boxing parameter that bounds the payment of the scoring rule $S$. Specifically, we assume $\| S \|_{\infty} < B_S$.
>
> **Q5:** How computationally intense is the solving of the LP programming in UCB-LP?
>
> **A5:** Thank you for your question. Our UCB-LP involves parameters and constraints that are polynomial in the number of arms $K$ and the number of belief $M$. Therefore, the LP can be solved in polynomial time w.r.t. $K$ and $M$. We also note that a similar LP was solved by Cacciamani (2023) (see their Eq (2)),  who provided a discussion on its computational complexity in Corollary 3.2, further supporting the tractability of our approach.
>
> **Q6:** Line 310 ... are happy about.
>
> **A6:** Thank you very much for your question. Our paper primarily focuses on the pure exploration setting, which is analogous to the pure exploration problem in the bandit literature [Lattimore (2020)]. In this context, we typically do not consider the regret incurred during the exploration phase. That said, there has been published work in the bandit literature on minimizing regret while identifying the best arm. We view this (minimizing regret and identifying the best scoring rule) as a valuable direction for future research.
>
> **Reference:**
>
> Chen (2023). Learning to incentivize information acquisition.
>
> Cacciamani (2023). Online information acquisition: Hiring multiple agents.
>
> Lattimore (2020). Bandit algorithms.
>
> Zhu (2022). The sample complexity of online contract design.
>
> Scheid (2024). Incentivized learning in principal-agent bandit games.

---

> > ### Comment · Reviewer_ksDs · 2025-04-03
> >
> > I thank the authors for their answer. Given the different feedbacks and authors' answer, I am sure the clarity of the paper will be improved in its revised version and thus decide to raise my score. However, I am still unsure of the relevance of this **underlying state** setting: how does it really make learning harder/different?

---

> > > ### Author Response · Authors · 2025-04-04
> > >
> > > We sincerely thank the reviewer for the positive feedback. We will revise the paper accordingly to improve its clarity. Below, we provide some intuition on why the presence of an underlying state increases the difficulty of the learning problem.
> > >
> > > In the setting of Scheid et al. (2024), the principal's objective is relatively straightforward: to incentivize the agent to take a good action, as the agent’s action directly determines the principal’s utility. In contrast, our setting introduces an added layer of complexity due to the presence of an underlying state that is unobservable to the principal but directly impacts the agent’s utility. The principal must rely on the agent’s report to derive the information about the state. Consequently, maximizing utility in our setting requires addressing two intertwined challenges: it must incentivize the agent to (1) take the optimal action (e.g., exerting high effort in research so that it can derive high quality information of the state), and (2) truthfully report its belief about the underlying state.
> > >
> > > Consequently, while the payment rule in Scheid et al. (2024) can be designed solely based on the agent’s action, the scoring rule in our setting instead depends on the agent’s report. This introduces another layer of difficulty: the scoring rule must not only incentivize the agent to take the optimal action, but also to truthfully report its belief. Hence, identifying the optimal scoring rule in our setting demands a more nuanced and sophisticated analysis than that in Scheid et al. (2024).
> > >
> > > Once again, we sincerely thank the reviewers for their valuable feedback and constructive questions and suggestions.

---

### Official Review · Reviewer_XcZh · 2025-03-13

**Overall Recommendation:** 3

**Summary:**

The paper studies how to learn scoring rules within the principal-agent problem for information acquisition. In particular, the authors study how to identify approximately optimal scoring rules with high probability through interaction with the environment. They design two algorithms depending on whether the setting is with fixed confidence or fixed budget. The first algorithm guarantees both an instance-dependent and an instance-independent bound with $O(1/\epsilon^2)$ samples, where $\epsilon$ is the required accuracy. This improves upon the $O(1/\epsilon^3)$ bound of Chen et al. (2023).  The second algorithm fixes the number of samples (budget) and identifies approximately optimal scoring rules.

**Claims And Evidence:**

Yes.

**Essential References Not Discussed:**

None, although a more comprehensive comparison with Cacciamani et al. (2023) could be beneficial. Indeed, since it generalizes the problem studied in the paper, their techniques can in principle be applied to the paper setting.

**Experimental Designs Or Analyses:**

Not applicable.

**Methods And Evaluation Criteria:**

Not applicable.

**Other Comments Or Suggestions:**

At line 173 in the "fixed budget" paragraph it is unclear which are the inputs of the problem (I guess T) and which are the outputs (I guess $\epsilon$ and $\delta$).

**Other Strengths And Weaknesses:**

The problem of learning how to incentivize information acquisition is interesting. The paper improves upon the guarantees of Chen et al. (2023).

The authors discuss extensively the relatrion between their work and Chen et al.. However, it remains unclear how novel the presented algorithm is since the structure is very similar to Chen et al. It would be interesting to better highlight the algorithmic contributions (except with setting parameters differently), and why the algorithm of Chen et al. fails to achieve optimal bounds.

I feel that also a more extensive comparison with Cacciamani et al. is required. Indeed, their work can be applied to your problem. Since the algorithm of Cacciamani et al. has an explore then commit flavor and achieve $T^{2/3}$ regret, I would not be surprised if it directly implies a $1/\epsilon^2$ sample complexity bound.

**Questions For Authors:**

Why are the algorithms of Chen et al. and Cacciamani et al. suboptimal for sample complexity problems?

Why is the dependency on $1/\epsilon^2$ required also in instance-dependent bounds?

**Relation To Broader Scientific Literature:**

The paper studies problems related to best arm identification for information acquisition. Previous works focus on the related problem of regret minimization.

**Theoretical Claims:**

I didn't check the appendix.

---

> ### Author Rebuttal · Authors · 2025-04-01
>
> We sincerely thank you for your insightful comments and questions, which have helped us improve the quality of our work.
>
> **Q1:** Why are the algorithms of Chen et al. and Cacciamani et al. suboptimal for sample complexity problems?
>
> **A1:** Thank you for the question.
>
> The primary reason for the suboptimal sample complexity in Chen (2023), aside from choosing an inappropriate parameter, is the adoption of a suboptimal termination criterion. Specifically, Chen (2023) employs the breaking rule originally introduced by Jin (2018), which terminates the algorithm after $\tilde{O}(\varepsilon^{-6} K^6 C_{\mathcal{O}}^3)$ iterations and selects the scoring rule identified in the final round as the best estimate. This termination strategy is inherently inefficient because it intuitively demands a large number of samples to ensure the accuracy of the identified scoring rule. In contrast, our proposed breaking rule (line 15 in Algorithm 1) and corresponding decision rule (line 16 in Algorithm 1) enable a more efficient evaluation of whether a scoring rule satisfies the necessary conditions. Consequently, our approach achieves significantly improved sample complexity.
>
> For Cacciamani (2023). We believe Cacciamani’s ETC algorithm can achieve a sample complexity bound of $K/\epsilon^2$ by carefully selecting the exploration phase based on the $\epsilon$. However, obtaining an instance-dependent sample complexity bound from ETC-based methods is challenging due to the necessity of setting the exploration phase length in advance, without prior knowledge of the actual instance-specific reward gaps or problem complexity. To our knowledge, there are currently no known ETC-based fixed confidence best arm identification algorithms in MAB problems that achieve instance-dependent complexity results.
>
> In summary, our algorithm achieves near-optimal instance-dependent sample complexity due to: (1) the selection of appropriate parameters, (2) the design of suitable breaking and decision rules, and (3) the adoption of an effective algorithm structure (UCB).
>
> **Q2:** At line 173 in the "fixed budget" paragraph it is unclear which are the inputs of the problem (I guess $T$) and which are the outputs (I guess $\epsilon$ and $\delta$).
>
> **A2:** Thank you for your question. In the fixed budget setting, $T$ and $\epsilon$ are the inputs, while $\tilde{\delta}$ is the output (the smaller the $\tilde{\delta}$, the better the algorithm’s performance). We will clarify this in the revisednext version of our paper.
>
> **Q3:** Why is the dependency on $1/\epsilon^2$ required also in instance-dependent bounds?
>
> **A3:** Thank you for this insightful question.
>
> To find the near-optimal scoring rule, we must identify both the best arm $k^*$ and the set of scoring rules that can trigger the best arm, denoted by $\mathcal{V}_{{k^*}}$. Intuitively, the dependency on $1/\Delta_i^2$ arises from identifying the best arm, whereas the dependency on $1/\epsilon^2$ emerges because, even after identifying the best arm, additional samples are required to estimate the set $\mathcal{V}{k^*}$ at an $\epsilon$-accurate level. Only then can we determine a scoring rule that meets the required accuracy.
>
> **Reference:**
>
> Chen, S., Wu, J., Wu, Y., & Yang, Z. (2023, July). Learning to incentivize information acquisition: Proper scoring rules meet principal-agent model. In International Conference on Machine Learning (pp. 5194-5218). PMLR.
>
> Cacciamani, F., Castiglioni, M., & Gatti, N. (2023). Online information acquisition: Hiring multiple agents. arXiv preprint arXiv:2307.06210.

---

### Official Review · Reviewer_DrTq · 2025-03-15

**Overall Recommendation:** 3

**Summary:**

The paper addresses the Best Scoring Rule Identification problem, proposing two algorithms: OIAFC (Online Information Acquisition Fixed Confidence) and OIAFB (Online Information Acquisition Fixed Budget). These algorithms aim to identify the optimal scoring rule through online learning. The key contributions are the introduction of a novel method to balance exploration and exploitation, the use of conservative scoring rules, and the design of efficient stopping and decision rules. The paper provides theoretical upper bounds for the sample complexity of these algorithms, showing that they can identify an estimated optimal scoring rule with high confidence, achieving instance-dependent and instance-independent bounds. The algorithms are evaluated in both fixed-confidence and fixed-budget settings, with an emphasis on improving sample efficiency compared to previous methods.

**Claims And Evidence:**

The claims made in the submission are generally supported by theoretical evidence. However, a few claims could benefit from more empirical or practical validation:
- Sample Complexity Bounds: While the paper presents theoretical upper bounds for sample complexity (both instance-dependent and independent), these bounds are derived under certain assumptions (e.g., the agent's behavior and the confidence levels). These claims could be seen as problematic if not empirically validated, especially since sample complexity is highly dependent on specific real-world conditions that are not fully addressed in the paper.
- The claim that forced exploration via binary search is efficient is supported theoretically, but the practical impact on performance (e.g., in terms of time or sample efficiency) is not clearly demonstrated. Empirical evidence or simulation results would help verify the actual effectiveness of this approach in practice.
- While the paper compares its results to prior work (like Chen et al., 2023), the paper primarily focuses on theoretical performance without empirical validation or concrete case studies. This leaves the claim that the proposed method is more efficient somewhat unsubstantiated outside the theoretical context.

**Essential References Not Discussed:**

The paper cites important works like Chen et al. (2023) and Jin et al. (2018), below lists additional papers that might provide related works in the broader best-arm identification and exploration-exploitation trade-off literature. Specifically:
- the work of Auer et al. (2002) on UCB (Upper Confidence Bound) algorithms is foundational and could help provide context for understanding the theoretical underpinnings of the proposed algorithms.
- The paper mentions fixed-budget and fixed-confidence settings, Kalyanakrishnan et al. (2012) might also be relevant to provide a more comprehensive understanding of multi-armed bandit algorithms under these constraints.
- While the paper introduces a more efficient exploration strategy, Lattimore and Szepesvári (2018) might also be relevant.

**Experimental Designs Or Analyses:**

no experimental designs in the paper.

**Methods And Evaluation Criteria:**

The OIAFC and OIAFB algorithms are well-suited to the problem as they are designed to identify the best scoring rule efficiently under both fixed-confidence and fixed-budget settings. These algorithms take into account the reward gap and problem complexity, which are critical for addressing the exploration-exploitation trade-off inherent in the multi-armed bandit framework. The use of conservative scoring rules, forced exploration, and binary search strategies are appropriate for ensuring that the best arm is identified with minimal sample complexity.

The evaluation criteria, specifically the sample complexity bounds (both instance-dependent and independent), are appropriate for measuring the efficiency of the proposed methods. The focus on deriving upper bounds for sample complexity is standard in MAB research, allowing for a theoretical assessment of the algorithm's performance. Additionally, the use of (ϵ, δ)-optimality as a measure of success aligns with common practices in bandit theory.

While the theoretical evaluation is solid, the absence of empirical results or benchmarks (e.g., performance on real or simulated datasets) makes it harder to assess the practical utility of the methods in real-world applications. Including empirical validation against standard MAB benchmarks would strengthen the claim that these methods are practically applicable.

**Other Comments Or Suggestions:**

NA

**Other Strengths And Weaknesses:**

Strengths:
- The paper introduces a novel algorithm (OIAFC) for best scoring rule identification, combining existing bandit methods with a conservative scoring rule to improve exploration efficiency. This approach is an interesting extension of prior work.
- The theoretical contributions are impactful, offering a refined understanding of sample complexity in the best arm identification problem, with applications to optimal scoring rule determination.
- The writing is generally clear, especially in explaining the algorithm’s workings and key definitions.

Weaknesses:
While the theoretical analysis is strong, there is little practical evaluation or comparison with state-of-the-art algorithms, which weakens the paper's applicability in real-world scenarios.

**Questions For Authors:**

1. Can you provide more detailed explanations or examples of how the reward gap and problem complexity are calculated in practical scenarios?

2. Can you elaborate on how the selection of the parameters alpha and beta affects the performance of the algorithm, especially in terms of sample complexity and exploration vs. exploitation balance? also the theoretical results mention upper bounds for the sample complexity, but it’s unclear how these bounds behave in practice with varying values of M, K, and the other parameters.

**Relation To Broader Scientific Literature:**

The key contributions of the paper—specifically the OIAFC and OIAFB algorithms—build on existing work in the Multi-Armed Bandit (MAB) literature, particularly in the context of best arm identification and optimal scoring rule problems. The approach extends prior work on bandit algorithms by introducing a more nuanced trade-off between exploration and exploitation, which improves the sample complexity bounds.

The paper distinguishes itself by introducing new parameters that fine-tune the balance between normal and forced exploration, contributing to a more efficient sample complexity, especially in instance-dependent settings. This contrasts with prior methods like those in Chen et al. (2023), which use simpler parameterization and result in higher sample complexity. Additionally, the comparison to Jin et al. (2018) highlights the inefficiencies of their naive breaking rule, further demonstrating the novelty and practical improvements of the proposed methods. In summary, the paper builds on established MAB and best arm identification frameworks but introduces more refined techniques for better sample complexity, offering an improvement over previous approaches.

**Theoretical Claims:**

High-level but did not follow the details and appendix. Based on the main submission, the proofs referenced in Lemmas, and Theorem 1 seem logically consistent with standard practices in multi-armed bandit theory, particularly around sample complexity bounds and exploration strategies, but admit for not detail following every steps in mathematical proofs.

---

> ### Author Rebuttal · Authors · 2025-04-01
>
> We sincerely thank you for your comments and questions.
>
> **About the references:**
>
> Thank you for pointing out these relevant references. We will include the three suggested papers in our revised manuscript. We greatly appreciate your helpful suggestion.
>
> **About the Experiments:**
>
> Thank you for your suggestion. In response, we have conducted additional experiments to validate the effectiveness of our proposed algorithm OIAFC.Thank you for your suggestion. We present several experimental results here to demonstrate the effectiveness of our algorithm OIAFC.
>
> **Experiment 1:**
> We set $B_S = 1$, the number of actions $K = 3$, the number of beliefs $M = 3$, and the number of states = 2.
> We vary the minimum reward gap between the best arm and the sub-optimal arm from 0.1 to 0.5 by adjusting $q_1,...,q_K$ and $c_1,...,c_K$.  We set $\delta = 0.05$ and $\epsilon = 0.1$. The results are presented in Table 1.
>
> **Experiment 2:**
> We fix the minimum reward gap at 0.2 and vary both the number of arms and the number of beliefs simultaneously from $\{2, 4, 6, 8, 10\}$, i.e., the number of arms equals the number of beliefs in each setting.
> All other settings are the same as in Experiment 1. The results are presented in Table 2.
>
> Each data point in the table is the averaged over 10 independent runs.
>
> **Table 1:**
> | Reward gap | Samples   |
> |------------|-----------|
> | 0.5        | 8909.4    |
> | 0.4        | 13972.5   |
> | 0.3        | 46331.3   |
> | 0.2        | 69542.0   |
> | 0.1        | 117963.8  |
>
> **Table 2:**
> | Arm and belief number | Samples   |
> |-----------------------|-----------|
> | 2                     | 54732.6   |
> | 4                     | 77321.5   |
> | 6                     | 89125.4   |
> | 8                     | 10052.9   |
> | 10                    | 113592.4  |
>
> As shown in Tables 1 and 2, the sample complexity increases as the reward gap decreases and as the number of arms and beliefs grows. These trends are consistent with theoretical expectations, since smaller gaps and larger action spaces make accurate identification more challenging.
>
> **Q1**：Can you provide more detailed explanations or examples of how the reward gap and problem complexity are calculated in practical scenarios?
>
> **A1**: Thank you for your question. We would like to clarify that our algorithm does not require knowledge of the reward gap or the instance-dependent problem complexity. These quantities—such as the reward gap and instance-dependent problem complexity—are unknown to the algorithm and are not used as inputs. Their role is purely theoretical, as they only appear in the analysis of sample complexity upper bounds. This is analogous to the setting in pure exploration bandit problems].
>
> **Q2：** Can you elaborate on how the selection of the parameters alpha and beta affects the performance of the algorithm, especially in terms of sample complexity and exploration vs. exploitation balance?
>
> **A2:** Thank you very much for your thoughtful question. Since our work focuses on the pure exploration setting, our algorithm is not concerned with the exploration–exploitation trade-off as in the regret minimization setting. The parameter $\alpha$ in our algorithm is used to balance the trade-off between exploration to identify the best scoring rule and exploration to estimate the feasible region $\mathcal{V}_k$ as shown in line 10 of Algorithm 1. The parameter $\beta$ is used in the breaking rule in line 15 to determine whether the optimal scoring rule has been attained with high probability.
>
> Specifically, if $\alpha$ is relatively large, the algorithm is more inclined to prompt the agent to choose the action we wish it to take, which reduces the number of binary searches required. For example, in the extreme case where $\alpha_k^t = 1$ for all $t$, the OIAFC algorithm would directly present the action-informed oracle of arm $k$, and the agent respond with $k_t = k$. However, this could potentially lead to an increased simple regret $h(S^*) - h(S_t)$, requiring more regular rounds to estimate a suitable $\hat{S}^*$ that satisfies the feasibility condition. Conversely, if we choose $\alpha_k^t = 0$ for all $t$, due to the absence of accurate estimation for the feasible region $\mathcal{V}_k$, the agent may not return the desired arm. In this case, the algorithm needs to perform more binary searches to estimate $\mathcal{V}_k$.
>
> As for $\beta$, setting it too high may cause the algorithm to terminate prematurely with a small number of samples, but the estimated best scoring rule may not satisfy the required condition. On the other hand, if $\beta$ is set too low, the algorithm becomes overly conservative and continues sampling until the reward gap between the estimated and true best scoring rule is much smaller than the target threshold $\epsilon$, which may lead to significantly higher sample complexity than necessary.

---

### Official Review · Reviewer_aBfB · 2025-03-23

**Overall Recommendation:** 4

**Summary:**

This work considers optimal scoring rules for principal/agent problems in online settings, in two variants: with a fixed budget, or fixed confidence, where the principal is trying to make investment decisions based on the knowledge/actions of the agent. The utility of the principal is driven by the quality of the knowledge/actions of the agent.

Prior work in scoring rules usually focuses on offline settings, so generating optimal fixed scoring rule according to a metric like minimizing regret on the principals part. Chen et al [2023] had studied the online setting with fixed confidence target, with a large instance-independent sample complexity bound.

This work adds instance-dependent parameters to the algorithm and the analysis, which improves both actual performance and sample complexity bounds.

**Claims And Evidence:**

Yes, the claims made in the submission are supported by clear theoretical evidence and proofs.

**Essential References Not Discussed:**

None that I am aware of.

**Experimental Designs Or Analyses:**

N/A

**Methods And Evaluation Criteria:**

Yes, the theoretical methods make sense for the problem at hand.

**Other Comments Or Suggestions:**

n/a

**Other Strengths And Weaknesses:**

The paper is well written and consistently raises the higher-level conceptual impacts and meanings of different aspects, starting from succinctly covering the reason these bounds should exist: the large gap between performance in non-incentive-based online bandit problems and the online scoring rule from Chen et al. 2023.

**Questions For Authors:**

none

**Relation To Broader Scientific Literature:**

This work improves significantly on the literature for online best scoring rule identification, with improved performance and sample complexity bounds that brings bounds closer to online bandit sample complexity.

**Theoretical Claims:**

I did not verify proof correctness.

---

> ### Author Rebuttal · Authors · 2025-04-01
>
> Thank you for recognizing the novelty and importance of our work. We truly appreciate your encouraging feedback.

---

### Decision · Program_Chairs · 2025-05-01

**Decision:**

Accept (poster)

**Comment:**

The reviewers found the work appealing and were generally in its favor. While they raised some issues, they did not find them to be sufficiently problematic to submit a single negative opinion (but, also, they did not engage much in defending the paper).